# Distinct neural mechanisms of social orienting and mentalizing revealed by independent measures of neural and eye movement typicality

Michal Ramot [1]*, Catherine Walsh[1], Gabrielle Elise Reimann[1] & Alex Martin [1]

Extensive study of typically developing individuals and those on the autism spectrum has identified a large number of brain regions associated with our ability to navigate the social world. Although it is widely appreciated that this so-called "social brain" is composed of distinct, interacting systems, these component parts have yet to be clearly elucidated. Here we used measures of eye movement and neural typicality—based on the degree to which subjects deviated from the norm—while typically developing ($N = 62$) and individuals with autism ($N = 36$) watched a large battery of movies depicting social interactions. Our findings provide clear evidence for distinct, but overlapping, neural systems underpinning two major components of the "social brain," social orienting, and inferring the mental state of others.

[1] Laboratory of Brain and Cognition, National Institute of Mental Health, National Institutes of Health, Bethesda, MD 20892, USA. *email: michal.ramot@nih.gov

Movie viewing involves many complex mental tasks. These include, although certainly are not limited to, applying mechanisms of directed attention to select the most relevant information and understanding the behavior of the characters and predicting their future actions through mentalizing. In social, dynamic scenes, attentional selection (measured through eye movements) is driven not only by low-level visual features, but is preferentially modulated by social cues. Saliency maps rely on low-level features such as contrast, color, and motion to predict fixations[1–3]. Yet, the presence of faces in the scene is a better predictor of fixations than saliency maps[4,5] and orientation to faces is further enhanced in the presence of accompanying speech[6]. Other social cues, such as gaze direction, emotion, and touch are also better at predicting attentional focus than low-level visual features[7] and information derived from head orientation and body position on top of gaze direction have also been shown to strongly modulate social attention[8,9]. Response to such social cues is often referred to as social orienting.

Directors of Hollywood movies are particularly adept at manipulating the focus of our attention, using cinematic techniques to tightly control where viewers' attention is drawn[10–13]. However, attentional synchrony, even when anchored around social cues, need not be driven by higher-order cognition or mentalizing, as argued convincingly regarding the non-human primate literature[14]. Similarly, in humans, attentional synchrony seems to be dominated by transient visual and social cues, and is only very weakly modulated by higher-level comprehension of the narrative, or inferences based on the mental states of the characters in the scene, as is demonstrated by studies, which manipulated comprehension through temporal shuffling of scenes[15] or manipulation of available context[16,17].

However, movie experiences are generally robustly shared across viewers at higher cognitive levels as well. Previous studies have described widespread correlations in neural responses across individuals, extending well beyond perceptual regions into higher-level processing regions[18]. This neural synchrony, or neural typicality, measured by the inter-subject correlations (ISCs) of the neural response time course to the movie, has been shown to underlie shared subsequent memories for events in the movie[19], a shared interpretation of the narrative[20], and can even predict friendship[21]. Thus, there appears to be a distinction between the lower-order process of choosing what to attend to and higher-order processes involved in the complex interpretation of what we saw. In the framework of Movie viewing, similarity in eye movement patterns reflects similarity in mechanisms of social orientation, whereas similarity in social comprehension and mentalizing would only be reflected in the degree of similarity of the neural responses in the brain regions involved in these tasks and not in the eye movement patterns.

It is particularly interesting to consider participants with autism spectrum disorder (ASD) in the context of this disassociation between social orienting and higher-order comprehension/mentalizing. Social deficits and impairments in social processing are among the defining characteristics of ASD[22]. For adolescents and young adults on the high functioning end of the spectrum however, these manifest most consistently as deficits in complex mentalizing or theory of mind tasks, although these deficits can be subtle and difficult to probe experimentally[23–26]. These difficulties also extend to social orienting. High functioning adolescents and young adults with ASD exhibit aberrant social orienting as manifested by aberrant eye movements to faces and other social stimuli, although, again, differences can be subtle and are usually only apparent when examined with complex stimuli or sensitive metrics[27–29].

Movie viewing is an experimental environment uniquely suited to the study of the social brain. On top of the robust and widespread basis of shared responses to the movie, both behavioral and neural, there is a range of individual variance[30]. Eye movement patterns, reflecting attentional selection, vary across individuals[31], as do all aspects of their social comprehension, from basic understanding to empathy for the characters in the scene[32]. This provides an exceptional opportunity for uncovering links between the brain and behavior, whereas the complexity and depth of the social stimuli make it ideal for picking out subtle differences in high functioning ASD. Previous research has focused on correlating the typicality of neural responses during Movie viewing to behavior related directly to the movie in question, such as memory for specific scenes[33]. Differences in ISC between typically developing (TD) and ASD groups have been examined in only a handful of studies, mostly with very few participants and mixed results[34–36]. Similarly, although many previous studies have used movies to study the behavioral aspects of social orienting[37–39], the search for the neural correlates of social orienting has so far utilized only very simple, mostly static and schematic, social stimuli. Moreover, these studies have focused on probing gaze following, which is only one aspect of social cues[9]. These limitations have led to a partial, fragmented understanding of the neural structures underlying complex social orienting behavior, which we will term the social orienting network.

Here we sought to exploit the full capacity of the Movie viewing environment by expanding the analysis to include an independent measure of behavior, with eye movements recorded during a different, independent set of movie clips than that used for the functional magnetic resonance imaging (fMRI) session. This allowed us to make inferences, which generalize beyond our specific movie stimuli. We compared measures of typicality for eye movements (measured by distance from the average scan path), while watching movie clips outside the scanner with the typicality of neuronal responses derived from voxel-wise ISC, while watching a different movie during fMRI acquisition in a large cohort (62 TD and 36 high functioning participants with ASD). This allowed us to uncover the broad neural underpinnings of the social orienting network, providing a much more detailed and complete delineation of this network than previously described using simplistic stimuli. A group comparison of the typicality of neuronal responses in TD and ASD participants revealed a second, distinct network, which in a manner congruent with the above observations does not correlate with the eye movements, but instead corresponds to regions previously implicated in mentalizing and theory of the mind. Together, these findings present direct evidence and a comprehensive description of two fundamental components of the social brain.

## Results

Sixty-two TD participants (24 female) and 36 participants with ASD took part in this study. Of the TD group, 36 were matched to the ASD group, in terms of gender (all male), age, IQ, and motion (see Methods for more details). In analyses where the TD group is considered separately, the full TD dataset was used, whereas in analyses comparing or combining the two groups, only the matched TD subset was used. All participants completed a behavioral session outside the scanner, in which they watched 24 short (14 s) movie clips taken from popular Hollywood movies, while their eye movements were being recorded. Three ASD participants and two TD participants did not achieve adequate calibration and were removed from the eye-tracking portion of the analysis. These movie clips were chosen in a separate pilot study from a larger set of 60 movie clips, for eliciting the most consistent viewing patterns across subjects. Immediately following the behavioral session, participants took part in an

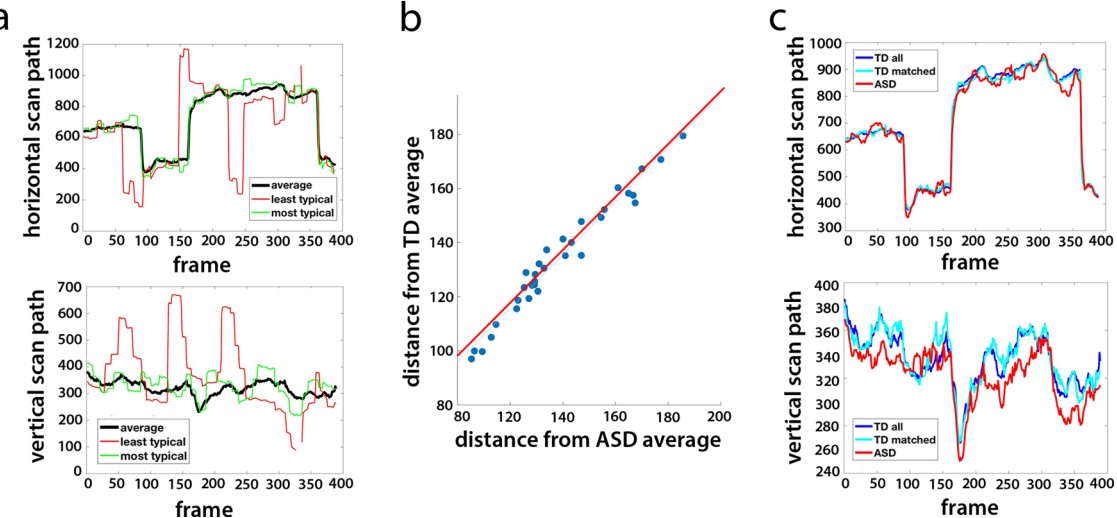

**Fig. 1 Eye movement typicality for TD and ASD participants. a** An example average scan path (black) along the horizontal and vertical dimensions, for one sample movie clip. Overlaid are the scan paths for the most typical (green) and least typical (red) TD participants. **b** The correlation between the typicality (shown as a distance measure) for each ASD participant (averaged across movies) when compared with the average of all other ASD participants (x axis) vs. the average typicality when compared with the average of all matched TD participants (y axis). Each dot denotes one ASD participant. **c** The average horizontal and vertical scan paths for the same movie as that shown in **a** for all TD participants (blue), the matched TD participants only (cyan), and all ASD participants (red).

fMRI scan session without eye tracking, during which they watched a 9.5 min clip taken from a different movie. All movie clips outside and inside the scanner depicted social scenes, with interactions between at least two characters and were presented with sound (see Methods). In addition, informant versions of the Social Responsiveness Scale (SRS) measure were obtained from the parent or guardian for the ASD participants.

**Eye movement typicality.** Even within these carefully selected movie clips, there was a range of individual eye movements, with some participants having more typical viewing patterns than others. For the TD group, we quantified the typicality of eye movements for each participant and each movie by calculating the Euclidean distance of their eye movements from the mean scan path of all other participants for each frame, averaging across all frames of that movie. This method gave us a distance measure of how similar that participant's viewing pattern was to the average viewing pattern of all other TD participants, per movie. Figura 1a shows an example of the mean scan path of all TD participants for one movie clip, with the eye movements of the most typical and least typical participants plotted in green and red, respectively. The average eye movement typicality of each participant was defined as the average distance across all 24 movie clips, with an inverse relationship between the two, so that the smaller the distance measure, the more typical the eye movements. This typicality metric served as a measure of the degree to which individual participants' eye movements differed from the group norm.

For the ASD group, we calculated two measures of typicality for each participant: one by quantifying the distance from the mean of all other participants in the ASD group and the other by quantifying the distance from the mean of all the matched TD participants. The two measures were nearly identical, with a correlation of $r = 0.99$ across participants between the average typicality when compared with the others in the ASD group and the average typicality when compared with the TD group (Fig. 1b). For individual movies, this correlation ranged between $r = 0.93$ and $r = 0.995$. This is in line with our finding that at the

group level, the average scan paths were very tightly coupled across groups—the average ASD scan paths for each movie to the average scan paths of both the matched TD subset and the full TD group were always more similar to each other than the average TD scan path was to the most typical TD participant, and correlations along the horizontal scan path were >0.88 for all movies. An example for the average scan paths for the two groups is shown in Fig. 1c. Given the high correlation between the two measures, we decided to henceforth define the eye movement typicality for the ASD group as the Euclidean distance from the mean scan path of the TD group, as this more obviously represents "typical" Movie viewing and social orienting at the population level.

**Stability of eye movement typicality.** To test whether this measure of eye movement typicality is a robust and stable individual subject trait, we first divided the 24 movie clips into 2 sets (odd and even) and calculated the mean typicality for each participant for each set of 12 movies. We next correlated this mean typicality between the two movie sets across participants, separately for the TD and ASD group. Figure 2a shows this correlation between the two sets of data, with the TD participants plotted in blue ($r = 0.78$, $p = 1.9 \times 10^{-13}$ for all TD participants and $r = 0.70$, $p = 2.8 \times 10^{-6}$ for the matched controls) and the ASD participants plotted in red ($r = 0.73$, $p = 8.5 \times 10^{-7}$). Finally, we repeated 10,000 iterations of this analysis, randomly dividing the movies into two sets each time, and calculated the mean correlation between the two data sets. To gauge the likelihood of randomly getting such correlations between the two halves of the data, we carried out a permutation test, randomly shuffling the subject labels for each iteration. The unshuffled mean correlations across iterations for all TD participants ($r = 0.73$) for the matched subset of TD participants ($r = 0.67$) and for the ASD participants ($r = 0.75$) were all entirely outside the random distribution (Fig. 2b). To rule out the possibility that this stability in the typicality measure was a spurious result of a consistent calibration shift at the individual level, we repeated this analysis after demeaning the data per participant and recalculated all the

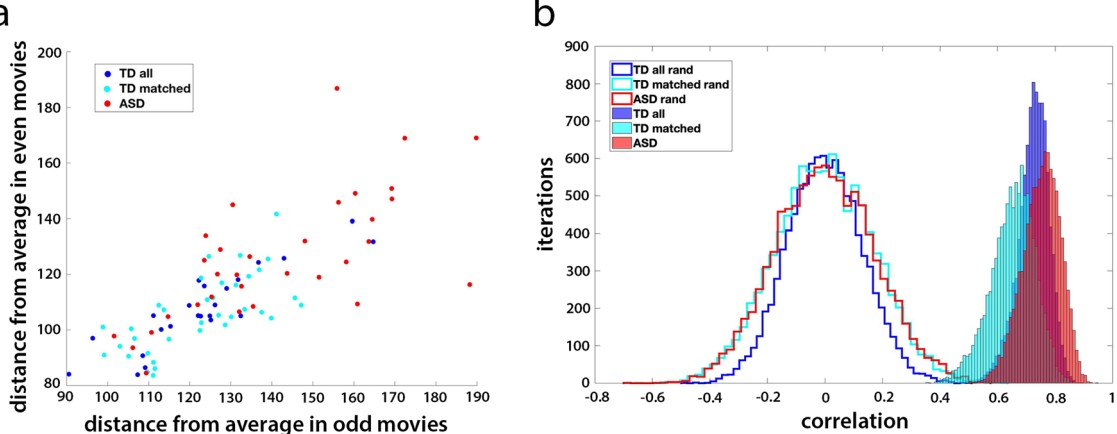

**Fig. 2 Stability of eye movement typicality. a** The mean typicality (shown as a distance measure) for each of the TD and ASD participants, averaged across all odd movies (x axis) vs. the mean typicality for each participant averaged across the even movies. Matched TD group participants shown in cyan, the remaining TD participants shown in blue, and ASD participants in red. **b** The random distribution centered around zero of the correlation between halves of the movie data across 10,000 permutations of different two-way splits for those three groups (blue, cyan, and red lines, respectively) vs. the true distribution (blue, cyan, and red histograms).

distances. The unshuffled mean correlations across iterations were very similar to before: $r = 0.71$ for all TD participants, $r = 0.63$ for the matched subset, and $r = 0.8$ for the ASD participants.

Despite the high correlations between the average scan paths of the two groups, TD participants had significantly more typical eye movements than their ASD counterparts (two sample two-tailed *t*-test of the matched TD vs. ASD participants, $p = 6.57 \times 10^{-6}$). Both groups exhibited a wide range of individual differences in eye movements (note the range of the distance measure across participants; Fig. 2), but the variance in typicality of the ASD group was significantly higher than in the matched TD subset (533 and 163, respectively, F stat 3.42, significant at $p = 6.57 \times 10^{-4}$), pointing to a wider range of behavior within the ASD group. This variance in behavior within the ASD group was not explained by social impairment, although, as measured by the SRS ($r = -0.14$, $p = 0.45$). Together, these results reveal the existence of a typical, "ideal" scan path for these movie clips, which is the same for TD and ASD participants. The difference between the TD and ASD groups is driven by increased variance and increased deviation from the same typical scan path in the ASD group.

**Neural typicality**. To assess the typicality of the neural responses for each participant during Movie viewing, we analyzed the fMRI data acquired, while participants were watching a different 9.5 min movie inside the scanner. We calculated the correlation of the time course of each voxel to the average time course of that voxel for all the other participants, giving us a measure of how typical (i.e., similar to the average) the neural responses to the movie were for that participant, per voxel. The map in Fig. 3 shows the average typicality of each voxel, defined as the mean typicality for that voxel across all TD participants. High typicality values indicate a high level of correlations across individuals in the activity of that voxel during the movie. High levels of these ISCs, while watching an engaging movie, spanned large areas of the cortex, including but not limited to sensory regions, spreading into many regions of association cortex, although sensory regions tended to be the most highly correlated between subjects. This is in accordance with several previous studies of ISC during Movie viewing[18,40,41]. It is noteworthy that due to scanning parameter constraints, the field of view did not cover the entire brain, with some areas primarily in motor cortex missing coverage. In

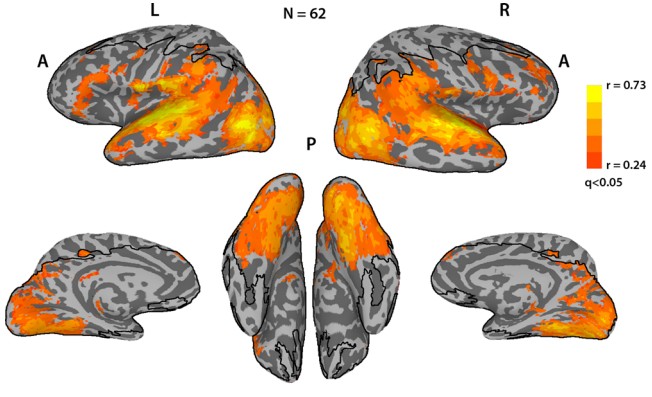

**Fig. 3 Average neural typicality.** Average neural typicality (ISC) across the entire brain for the TD subjects, thresholded at $q < 0.05$, FDR corrected. Black lines delineate the field of view; voxels outside this boundary were not imaged or were removed from the analysis for poor temporal signal-to-noise ratio (tSNR).

addition, some voxels were removed from the analysis for failing to meet minimal signal-to-noise requirements (see Methods).

We carried out the same analysis for the ASD group and, as with the eye movements, calculated neural typicality for each participant for each voxel to both the average time course of that voxel for all other ASD participants, and to the average time course of all the matched TD participants. As with the eye movements, these measures were very tightly correlated (mean correlation across participants was $r = 0.92$, calculated across voxels for each participant, and then averaged across participants), so we will henceforth define the neural typicality of the ASD group in relation to the TD average, for similar reasons as above. Supplementary Fig. 1 depicts the average neural typicality of the ASD group in relation to the TD average.

**Correlation between eye movement typicality and neural typicality**. To search for the neural underpinnings of social orienting, we sought to combine these two independent measures of the brain and behavior by conducting a whole brain search for voxels in which there was a correlation across participants of the typicality of the neural responses to the movie and the typicality

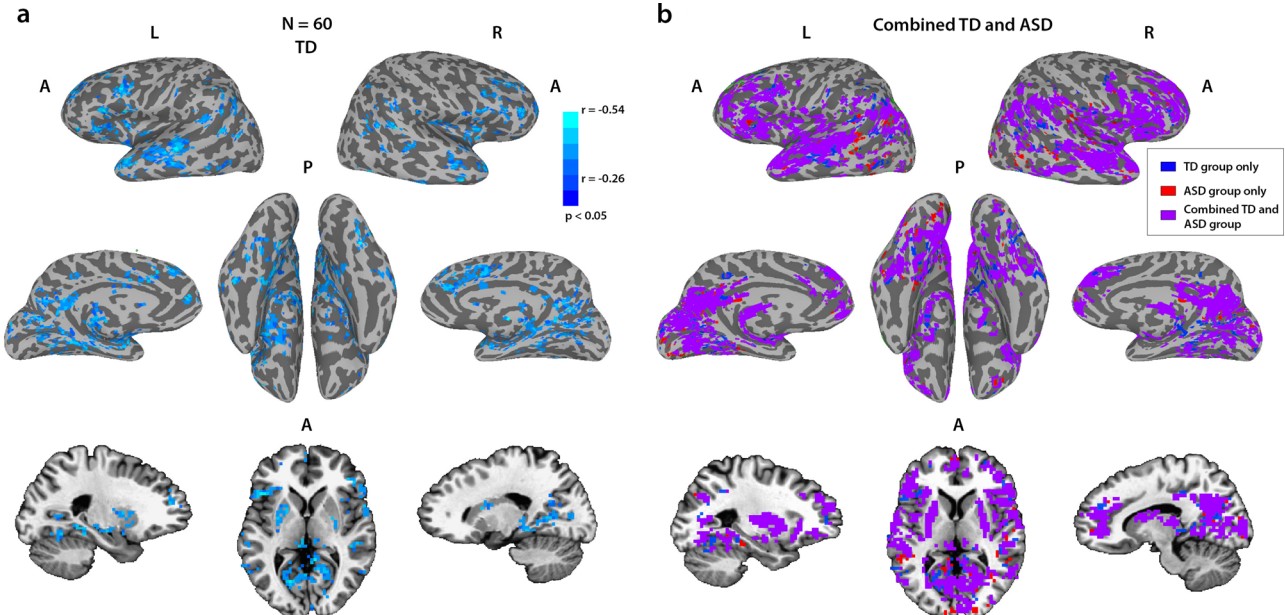

**Fig. 4 Correlations between neural typicality and eye movement typicality. a** Correlations between eye movement typicality and neural typicality for the full TD group at a corrected threshold of $p < 0.05$, corrected through cluster size permutation testing. **b** All voxels whose neural typicality was significantly correlated with eye movement typicality, for the combined group of the ASD participants and their matched controls (purple, $N = 69$), with voxels that were only correlated for the corresponding analysis of the matched TD group ($N = 36$) in blue and voxels that were only correlated for the corresponding analysis of the ASD group ($N = 33$) in red. It is noteworthy that the combined analysis is more widespread than either analysis separately.

of the eye movement patterns during the short movie clips shown outside the scanner. It is noteworthy that the eye movement typicality measure as we have defined it is in fact a distance measure, meaning that the greater it is, the less typical the eye movements. Voxels in which there was a significant anti-correlation (defined by calculating the $p$-value of the correlation coefficient from the appropriate $t$-values, given the sample size, at a threshold of $p < 0.05$) between the two measures are those in which the more typical (similar to the average) the neural response to the movie, the more typical the eye movements of that participant were to the short movies outside the scanner.

Figure 4a shows the results of this analysis for the TD group, revealing multiple regions associated with social and language processes, including superior temporal sulcus (STS), inferior frontal gyrus (IFG), anterior insula, posterior and anterior cingulate cortex (PCC and ACC, respectively), medial prefrontal cortex (MPFC), and subcortically the hippocampus, putamen, and caudate, bilaterally, for which eye movement typicality is strongly correlated with neural typicality in response to a movie (corrected for multiple comparisons through a permutation-based cluster size correction, $p < 0.05$). The results of the same analysis correlating eye movement typicality with neural typicality for the ASD group are displayed in Supplementary Fig. 2a and are centered on very similar regions.

To directly test whether the same network, which we found in TD participants, also underlies social orienting in participants with ASD, we examined the correlation between eye movement typicality and neural typicality within the network defined by the TD group for the ASD group. We defined a mask of the voxels, which were found to be significant in the TD analysis and averaged the neural typicality within this mask for each of the ASD participants. We then correlated this average neural typicality value with the eye movement typicality value across all the ASD participants, and found a significant correlation ($r = -0.42$, $p = 0.01$). Supplementary Fig. 3 shows a scatter plot with these data, overlaid with a similar analysis for the entire TD group ($r = -0.56$, $p = 4.2 \times 10^{-6}$) and the matched TD group ($r =$

$-0.59$, $p = 1.6 \times 10^{-4}$). The high correlations for the TD groups are expected, as it is this correlation that was used to define the network. These data are shown together only to put the ASD data in context.

As the same network seemed to underlie social orienting in both TD and ASD participants, we carried out an additional analysis on the combined group of the ASD participants with the matched TD participants. Figure 4b shows the overlay of this analysis with the analyses of just the matched TD participants (blue) or just the ASD participant (red). Combining the groups (the ASD group with the matched TD group) gave more widespread correlations with the eye movements, and these overlapped with the TD-only and ASD-only analyses substantially, with only 14% of the voxels in those two analyses not contained within the combined group analysis. To ensure that these correlations across the combined TD and ASD group were not driven purely by group differences in eye movement typicality and/or neural typicality (see below), we carried out an additional analysis controlling for the mean group effects. In this analysis, we subtracted the group average for both eye movement typicality and neural typicality from each of the TD and ASD groups, before combining the two. The results are displayed in Supplementary Fig. 2b. When the variance accounted for by the group means is removed, the social orienting network revealed by this analysis is more limited than that seen in Fig. 4b and more similar to the results for the separate TD and ASD groups (Fig. 4a and Supplementary Fig. 2a).

**Group differences in neural typicality**. The eye movement analysis captured many similarities between the two groups and yet there are clearly differences in social processing between them. We hypothesized that social difficulties in the ASD group would translate to less typical processing in the relevant social brain regions and would therefore be reflected in reduced neural typicality in those areas. To test this, we carried out a group $t$-test on the neural typicality measure of the matched TD participants and

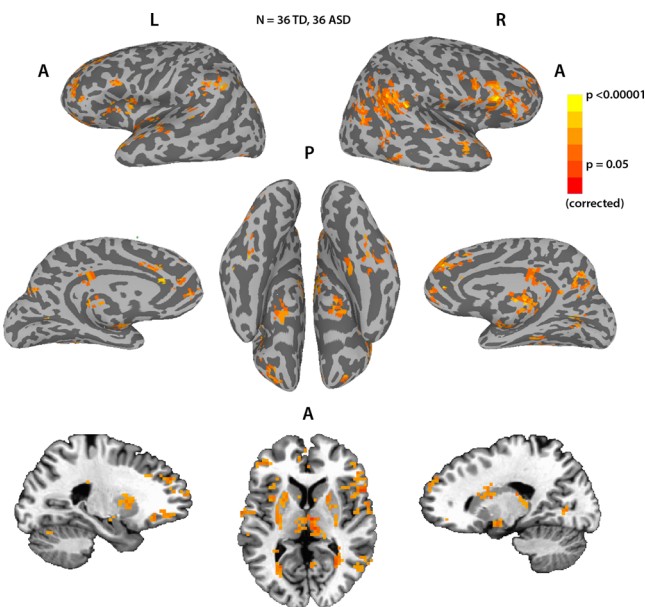

**Fig. 5 Group differences in neural typicality.** Voxels that had significantly greater neural typicality for the matched TD group than the ASD group. No voxels showed significant differences in the other direction. Corrected using cluster size permutation testing.

the ASD group, and the result is displayed in Fig. 5. Significant differences between the TD and ASD groups were found mainly in the right temporal parietal junction (TPJ) and middle temporal gyrus, and bilaterally in posterior STS, IFG, anterior insula, ACC, PCC, putamen, and caudate.

Surprisingly, many of the voxels in the social orienting network identified by the eye movement typicality analysis did not show significant differences in neural typicality as we would have predicted, considering the difficulty in social orienting that is also a hallmark of ASD. To test whether this was a result of a lack of sensitivity of the neural typicality measure, we examined the neural typicality group difference when averaging the neural typicality across the entire social orienting network, as defined by the TD group alone, and also as defined by the combined matched TD and ASD group (see Fig. 4). We found that even though there were no significant group differences in many of the individual voxels, the average neural typicality across the entire network was indeed significantly greater for the TD group both when using the network definition derived from the TD group alone ($p = 3.5 \times 10^{-4}$, two-tailed two sample $t$-test using the matched TD group) and when using the network definition derived from the combined group $p = 8.8 \times 10^{-5}$, two-tailed two sample $t$-test using the matched TD group). At the network level therefore, the social orienting network showed significantly greater neural typicality for the TD group compared with the ASD group.

**Two distinct networks**. So far, we have used two separate and very different analyses to identify two networks—the social orienting network identified through correlations of neural typicality with eye movement typicality (Fig. 4), and a second network, defined by group differences in neural typicality between the TD and ASD groups (Fig. 5). For simplicity, we will refer here to this second network as network 2. To examine the overlap between these two networks, we created a conjunction map of voxels belonging to just one of these analyses or to both, using a threshold of $p < 0.01$ and correcting for multiple comparisons

through cluster size permutation testing (Supplementary Fig. 4). Although there are some areas of overlap in STS, IFG, anterior insula, ACC, PCC, and putamen, other regions belong only to the social orienting network (anterior STS bilaterally, left pSTS and IFG, and most of ACC and PCC bilaterally), or are not correlated with eye movement typicality but show group differences in neural typicality, making them a part of network 2 (right TPJ, middle temporal gyrus, amygdala). To test whether the non-overlapping parts of these networks are functionally distinct, we examined whether the average correlation of each voxel to all the other voxels within the network was significantly greater than its average correlation to all the voxels in the other network. Results are displayed in Fig. 6.

There was a significant difference for the social orienting network, with correlations within the social orienting network significantly greater than correlations between the social orienting network and network 2 for both the TD and the ASD groups (Fig. 6a, b, paired two-tail $t$-test, $p = 5.4 \times 10^{12}$ and $p = 5.5 \times 10^{-13}$ for both groups). Correlations within network 2 were significantly more correlated within than across networks for both TD and ASD participants (Fig. 6e, f, $p = 4.4 \times 107^8$ for TD participants and $p = 0.036$ for ASD participants). Importantly, not only was there an overall significant bias for within vs. across-network correlations, but the subset of voxels showing significantly greater within than across-network correlations in the TD participants overlapped almost entirely with the subset of voxels in the ASD group, which were similarly more correlated within rather than across network, for both networks (99% overlap for voxels in the social orienting network (Fig. 6c, d) and 94% overlap for voxels in network 2 (Fig. 6g, h)). Figure 7 portrays the non-overlapping subset of the social orienting network and network 2 identified in Figs. 4 and 5 further limited to the voxels (identified in Fig. 6c, d, g, h), which show consistently greater within- than between-network correlations in both the TD and the ASD groups. There is an apparent laterality bias difference between the two networks, with the network 2 biased towards the right hemisphere (78% of voxels fall in the right hemisphere), whereas the social orienting network has a (weaker) left hemisphere bias, with 60% of voxels on the left.

To further examine the distinction between the networks, we tested whether the neural typicality within these two networks shown in Fig. 7 was differentially correlated with the SRS in the ASD group. This was indeed the case, with the neural typicality averaged across network 2 correlated to the SRS at $r = -0.35$, $p = 0.037$, whereas the neural typicality averaged across the social orienting network was not significantly correlated to the SRS at $r = -0.15$, $p = 0.38$. To test whether the correlation of the neural typicality difference network was significantly greater, we used the Steiger $z$-score test for the difference between two dependent correlations with one variable in common and the result was significant with a one-tailed test ($z = -1.9$, $p = 0.028$). It is noteworthy that these correlations are negative, with a higher (more impaired) SRS score correlating with lower neural typicality, as expected. The regions that overlapped between the social orienting network and network 2 were not significantly correlated to the SRS ($r = -0.22$, $p = 0.21$), but also not significantly different from the correlation of the non-overlapping subset of network 2 to the SRS ($z = 1.2$, one-tailed test $p = 0.11$).

## Discussion

Movie viewing evokes both shared neural responses, and shared behavior, in the form of eye movements orchestrated by carefully constructed visual, auditory, and social cues. Yet, there is individual variation in both behavior and neural responses. Evidence from previous studies suggests that neural typicality, or

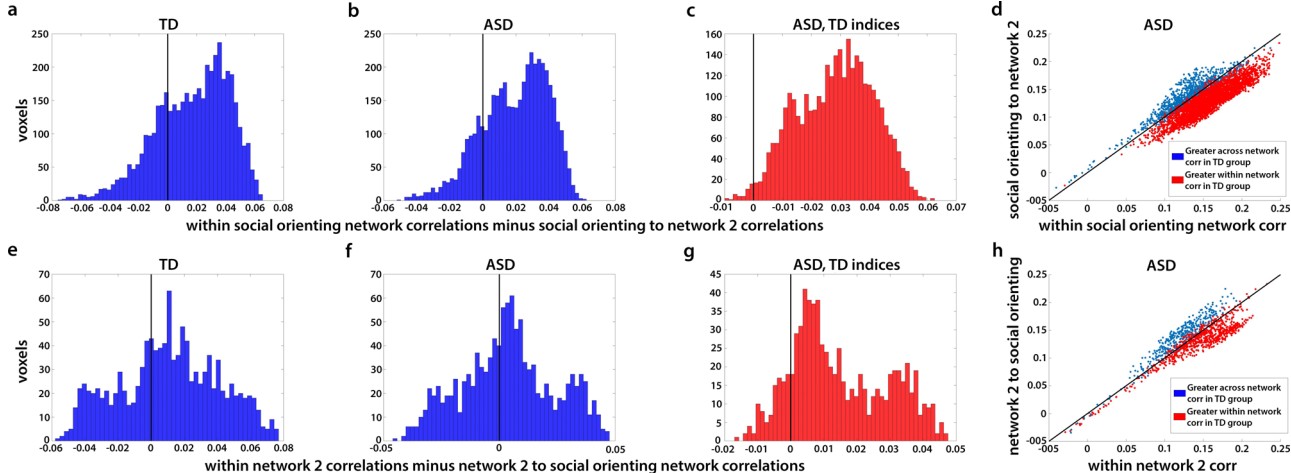

**Fig. 6 Distribution of within vs. across-network correlations for all non-overlapping voxels identified in the analyses portrayed in Fig. 4 (social orienting network) and Fig. 5 (network 2). a** Histograms shows the average correlation of each voxel in the social orienting network to all other voxels in the social orienting network, minus its average correlation to all voxels in network 2, averaged across all matched TD participants. Vertical black line at 0 denotes equal correlation, i.e., voxels are equally correlated to the other voxels within network as to voxels across network. Voxels to the left are more correlated across networks than within network (i.e., higher correlation to voxels in network 2), and voxels to the right are more correlated within network than across networks. **b** Same for the ASD group. Note the significant rightward shift in both these plots ($p = 5.4 \times 10^{12}$). **c** Distribution in the ASD group of all the voxels in the social orienting network, which were more correlated within the social orienting network than to the voxels in network 2 in the TD group, using the TD indices for these voxels. Ninety-nine percent of voxels with greater within vs. across-network correlations in the TD group were also more correlated within network in the ASD group. **d** Scatter plot of the same analysis shown in **b** and **c**. Each dot represents one voxel in the social orienting network; value on the x axis reflects average correlation to all other voxels in the social orienting network and value on the y axis reflects average correlation to all voxels in network 2, averaged across all ASD participants. Identity line (i.e., equal correlation to both networks) marked in black. Blue dots are voxels that were less correlated within than across network in the matched TD group, whereas red dots are voxels which had greater within than across-network correlations in the matched TD group. Note the almost complete correspondence across the two populations. Panels **e–h** show the same analyses as above, for correlations of voxels in network 2 to all other voxels in network 2 vs. their correlations to voxels in the social orienting network.

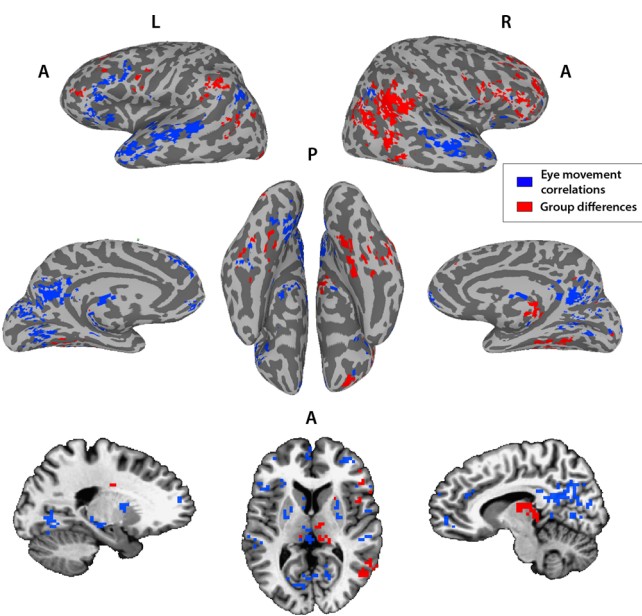

**Fig. 7 Two distinct networks derived from neural typicality group differences and eye movement correlations.** Overlay of the subset of voxels with significant correlations between neural typicality and eye movement typicality of the combined matched TD and ASD groups ($N = 36$ TD + 33 ASD, blue) and voxels showing significant group differences in neural typicality between the matched TD and ASD groups ($N1 = 36$ TD, $N2 = 36$ ASD, red), which also show greater within than across-network connectivity. Threshold set at $p < 0.01$.

correlations between subjects in their neural responses indicate shared processing or a shared experience relevant to the function of the correlated region. For example, studies have demonstrated that for participants viewing a movie, or listening to a narrative, the similarity of their interpretation of the story predicts the degree of neural similarity in regions involved in narrative interpretation, such as the fronto-parietal network and the default mode network[20]. Likewise, participants recalling the same events have been shown to have more neural similarity than participants recalling different events[19]. ASD participants have difficulties in both social orienting and higher-order mentalizing and social comprehension. These are likely the result of different processes, as previous studies have demonstrated a decoupling of high order comprehension/mentalizing during Movie viewing from eye movement patterns[15]. Using our measures of the correlation between eye movement typicality and neural typicality, and group differences between the TD and ASD groups in neural typicality, we therefore expected to find two different, partially overlapping networks. The correlation between the two typicality measures across participants was used to search for the network responsible for social orienting. As the neural typicality measure is an indicator of shared processing, we expected lower neural typicality in the ASD group in regions involved in the processes in which ASD participants differ from their TD counterparts, namely social orienting and higher-order social comprehension/mentalizing. Therefore, we expected to find group differences in neural typicality in the regions underlying these behaviors.

For the social orienting network, on top of previous research[27–29], our own data described here shows reduced typicality of eye movements in the ASD group compared with the TD group (Fig. 2) and this should be reflected in the typicality of neural processing. However, at the single voxel level, we did not find group differences in neural typicality in most voxels of the

social orienting network. Those differences were present at the network level though, as there was significantly lower neural typicality in the ASD group within the social orienting network as a whole, perhaps indicating a lack of sensitivity of the neural typicality group difference measure. The more robust effects, which could be seen at the single voxel level (Fig. 5) centered on regions which seemed to constitute a separate, distinct network, whose correspondence to the mentalizing network will be discussed below.

Eye movements have been used extensively to study social orienting[27,42,43], although to our knowledge this is the first time that an independent measure of typicality, using a different set of stimuli and based on the stability of eye movement typicality as a measurable individual trait, has been used to predict similarity of neural activity. The idea that this would be an informative measure is based on the notion that certain stimuli, especially those created explicitly to draw attention in very specific ways such as Hollywood movies, would have a "typical," or in a sense ideal scan path. Indeed, our movie clips were pre-selected in a pilot study (on a separate group of participants) for the consistency of the eye movement patterns they elicited. Despite this being an obvious oversimplification which disregards details and dynamics, the striking similarity between the mean scan path of each of our movie clips not only within TD participants but also between TD and ASD participants as is demonstrated in Fig. 1 indicates that such a stereotypical scan path for these clips exists.

Having established that there is a basis for considering typicality, we next investigated whether eye movement typicality would prove to be a stable individual trait, which is not specific to a particular stimulus or context (at least within the broader framework of social Movie viewing). This would be a prerequisite for using it as a marker for general social orienting abilities. The highly significant correlations of eye movement typicality across different movie clips (Fig. 2) pointed to this being a robust measure of social orienting at the level of the individual. This gave us an independent behavioral measure, with which to search for the neural correlates of social orienting.

Similar to the eye movements, the concept of neural typicality, more often referred to in the literature as ISCs, is based on the premise that neural responses across individuals will be similar when driven by a shared stimulus, in regions whose processing is stimulus related[40]. Figure 3 replicates the results of several studies, which have consistently shown significantly shared neural responses across these same cortical regions during Movie viewing[18,40,41]. A whole brain search for voxels whose neural typicality is strongly coupled to eye movement typicality across subjects was therefore a natural next step to revealing the neural substrate of social orienting.

The regions that exhibited a correlation between the two measures (Fig. 4) were very stable between the TD and ASD groups (Supplementary Figs. 2 and 3). Monkey electrophysiology, as well as lesion studies in both monkeys and humans, have all pointed to an important role for STS in gaze perception and social orienting[44–46]. Human neuroimaging studies have suggested that networks involved in gaze perception extend well beyond STS, including also ACC, MPFC, and hippocampus (see ref. 9 for a review). Recent studies have further expanded the networks involved in social orienting, with one study finding increased activation in STS, IFG, and the putamen when comparing gaze cues with non-social symbolic cues[47]. This study further found an interaction effect between groups (TD vs. ASD) and cue type (social vs. symbolic). These human neuroimaging studies are limited however by the very simplistic and specific nature of their stimuli (mostly cartoon gaze cues), which is perhaps why we were able to uncover a broader network linked to social orienting, which in our case is derived from a host of social cues that can be extracted from the naturalistic and dynamic movies. Please note however that we do not have full coverage of the brain and some areas that might also be involved in social orienting, most notably frontal and supplementary eye fields as well as some parietal regions along the intraparietal sulcus, are missing from our analysis (Fig. 3). These regions have previously been implicated in spatial attention tasks, and in particular attention to action perception, making them potentially relevant for social orienting[48]. However, frontal and supplementary eye fields are also directly involved in the control of eye movements, and it would therefore be difficult to decouple their role in eye movements from their role in social orienting. Other areas where we are missing coverage are mostly motor and are not implicated in either social orienting or mentalizing based on previous literature.

The eye movement analysis was intended to capture measures of social orienting, but it did not directly address the differences between the TD and ASD participants. To directly test for this, we examined the group differences in neural typicality between our TD and ASD participants (Fig. 5). There have been very few studies looking at differences in neural typicality or ISC between TD and ASD groups. The first[35] found differences in visual and auditory regions, but was very underpowered (with 12 ASD and 8 TD participants), whereas another found differences only between a subset of ASD participants[34]. A third study, though only slightly less underpowered ($N = 13$ in each group), used a full-length 67 min movie, which may have compensated for the low number of participants[36]. This study identified very similar regions to those found in our analysis. These regions have been shown in the past to be involved in mentalizing and in theory of mind[26,49–52]. Right TPJ in particular has long been thought to be fundamental for theory of mind[53–55], whereas left TPJ, MPFC, ACC, PCC, and IFG, have all been found to activate for various mentalizing and theory of mind tasks (see Schurz et al.[56] for a meta-analysis). In a study reporting differences between TD and ASD groups in a task, which required inferring intentionality from eye gaze, the same regions in right TPJ (referred to there as posterior STS) and middle temporal gyrus were identified as activating more in the TD group[57]. Amygdala on the other hand, is consistently found to be crucial for emotional processing, as well as for social/saliency perception, among its many roles[58–60]. The fact that significant neural typicality differences, at least at the individual voxel level, were found only for these regions, suggests that these regions are involved in the processing that is most atypical in ASD, which should correspond to the social tasks with which the ASD have the greatest difficulty.

The double disassociation between the two networks identified through the eye movement and neural typicality group difference analyses and our two behavioral measures (eye movement typicality and SRS), together with the consistency of greater within compared with across-network functional connectivity (Fig. 6), strongly point to the two sub networks, which also show this pattern of connectivity, being two distinct networks (Fig. 7). Considering the nature of the processing that takes place while watching social movies, the nature of the deficits in ASD, and the previous literature on the regions identified by this analysis, we hypothesize that this second network revealed by the neural typicality group difference analysis corresponds to the mentalizing network. We also found substantial overlap between the two networks using the broader definitions without the requirement of greater within than across-network connectivity, most notably in right IFG, MPFC, and PCC, and bilaterally in regions of putamen and the caudate (Supplementary Fig. 4). As the two functions of social orienting and mentalizing interact constantly, so must the networks that guide them, and it is perhaps in these regions of overlap that this neural interaction takes place.

## Methods

**Participants**. Thirty-six males aged 15–30 years (mean age = 20.7), who met the DSM-IV criteria for autistic disorder, an autism cutoff score for social symptoms on the Autism Diagnostic Interview–Revised and/or and ASD cutoff score from social and communication symptoms on the Autism Diagnostic Observation Schedule, all administered by a trained, research reliable clinician, were recruited for the experiment. In addition, 63 TD participants (24 female) aged 15–30 years (mean age = 22.05) were recruited. One was excluded from the analysis because of abnormal brain structure. Of the remaining 62, a subset of 36 males were chosen to match the ASD group, based on gender, age, IQ, and motion. Mean age for the matched TD group was 20.8 years (range 15–28). For the eye movement analyses, three participants from the ASD group and two from the TD group were excluded for failing to achieve adequate calibration on the eye tracker. All participants were right handed, and had normal or corrected to normal vision. IQ scores were measured by the Wechsler Abbreviated Scale of Intelligence, the Wechsler Adult Intelligence Scale-III, or the Wechsler Intelligence Scale for Children-IV. Full-scale IQ scores were all >94 and were matched between the ASD and the TD groups (mean IQ for the ASD participants was 108.7, std = 13.8; mean IQ for the matched TD participants was 110, std = 13.3). The experiment was approved by the NIMH Institutional Review Board (protocol 10-M-0027). Written informed consent was obtained from all participants or their guardians in the case of minors, in which case written assent was also obtained from the participants themselves.

**Eye-tracking setup**. Eye tracking was recorded with the Eyelink 1000 Plus. Participants' heads were stabilized using a chin and forehead rest, and eye gaze calibration was performed at the beginning of the viewing session for each participant. The movie clips were shown in a randomized order, with a brief 6 s pause in between successive clips, during which time a gray screen with a fixation cross was presented. The same screen and fixation were also presented before the first movie presentation. Movies were viewed full screen on a digital monitor with a 1920 × 1080 resolution, with a screen size of 20.5 × 12 inches. Eye-tracking data were sampled at 1000 Hz.

**Eye-tracking preprocessing and analysis**. Eye movement data was extracted for each movie clip separately. The first and last 500 ms were removed, blinks and missing (offscreen) data were ignored, and the data were despiked. Data were then down sampled from 1000 Hz to the frame rate at which the movies were presented (29.97 fps). This was then used to calculate the position of the eye on the screen for each frame, without identifying individual fixations. The average position for all (other) participants in the typicality analyses was calculated in the same manner.

**Movie stimuli**. An initial set of sixty 14 s movie clips was tested on an independent pilot set of 12 participants. The movie clips were then analyzed for the consistency of the eye movement patterns they elicited across these participants and the 24 most consistent movie clips were selected for the eye-tracking session outside the scanner. These movie clips were all taken from Hollywood movies—The Blind Side (6 clips), Goonies (4 clips), How To Lose a Guy in Ten Days (4 clips), The Italian Job (5 clips), and The Never-ending Story (5 clips), and were all 14 s long. There was ongoing dialogue between at least two characters in all these scenes, though characters were often not on the screen together. For the 9.5 min movie clip shown during fMRI acquisition, a scene from the Princess Bride was selected. This scene involved multiple characters on screen simultaneously, as well as continuous dialogue. All movies were presented with audio, using the built in speakers on the MacBook Pro for the movie clips shown during the eye-tracking portion of the experiment, and using the SilentScan headphone system from Avotec Inc. during fMRI scanning.

**Imaging data collection, MRI parameters, and preprocessing**. All scans were collected at the Functional Magnetic Resonance Imaging Core Facility on a 32-channel coil GE 3T (GE MR-750 3.0T) magnet and receive-only head coil. The scans included a 5 min structural scan (MPRAGE) for anatomical co-registration, with the following parameters: echo time (TE) = 2.7, flip angle = 12, bandwidth = 244.141, field of view (FOV) = 30 (256 × 256), slice thickness = 1.2, axial slices. Echo-planar imaging (EPI) scans were collected with the following parameters: repetition time (TR) = 2 s, Voxel size 3*3*3, Flip Angle: 60, multi-echo slice acquisition with three echoes, TE1 = 17.5 ms, TE2 = 35.3 ms, TE3 = 53.1 ms, Matrix = 72 × 72, slices: 28. Two hundred and eighty-five TRs were collected for the movie (9 min and 30 s). All scans used an accelerated acquisition (GE's ASSET) with a factor of 2, to prevent gradient overheating.

Post-hoc signal preprocessing was conducted in AFNI (Analysis of Functional Neuro-Images[61]. The first four EPI volumes from each run were removed to ensure remaining volumes were at magnetization steady state and remaining large transients were removed through a squashing function (AFNI's 3dDespike). Volumes were slice-time corrected and motion parameters were estimated with rigid body transformations (through AFNI's 3dVolreg function). Volumes were co-registered to the anatomical scan. The data were then entered to a Multi-Echo ICA analysis, as described in ref. [62], to further remove nuisance signals (e.g., hardware-induced artifacts and residual head motion). Briefly, this procedure utilizes the physical properties of blood oxygen level dependent (BOLD) and non-BOLD fluctuations,

namely the fact that although signal from BOLD sources increases linearly over echo times, signals from non-BOLD sources remain constant across echoes. This allows the removal of non-BOLD fluctuations (noise). The functional and anatomical data sets were co-registered using AFNI, and then transformed to Talairach space. Voxels with a temporal signal-to-noise ratio (tSNR) under 40 were removed from further analysis. tSNR was calculated by dividing the mean of the signal for each voxel by the temporal standard deviation, averaged across participants.

**Statistics and reproducibility**. All data were analyzed with in-house software written in MATLAB, as well as the AFNI software package. Data on the cortical surface were visualized with SUMA (Surface Mapping[63]). Eye movement typicality for each of the TD participants was calculated per movie per frame, by computing the Euclidean distance of their average recorded eye position per frame to the mean recorded eye position of all other participants (i.e., all N1 participants, not including the participant of interest) for each frame. These distances were then averaged for each participant across all frames of all movies, to obtain one typicality measure. Neural typicality was similarly assessed per participant, per voxel, by calculating the correlation of the time course of each voxel for that participant to the average time course of that voxel for all the other (N-1) participants. The correlation between the eye movement typicality and the neural typicality was calculated based on these two measures.

Two-tail *t*-tests were used for all *p*-values on correlations, unless otherwise stated. For the maps in Fig. 4, which were corrected through a permutation-based cluster size correction, we permuted the subject labels for the eye-tracking typicality for 10,000 iterations and correlated these with the neural typicality. The cluster threshold was defined for each threshold separately ($p < 0.05$, $p < 0.01$, $p < 0.005$, $p < 0.001$) as the largest cluster at that threshold at the 95th percentile across all iterations. Voxels were considered significant if they were significant in any of the corrected cluster sizes for the corresponding threshold, thus allowing the inclusion of both large clusters significant at lower thresholds, and smaller, highly significant clusters. This approach has been advised following the recent debate on cluster size corrections[64]. The same analysis was carried out for the results in Fig. 5, permuting the TD and ASD labels for 10,000 iterations. For Fig. 6a–d, we first calculated for each participant the correlations of each voxel within the social orienting network to every other voxel within the social orienting network, and averaged across all the correlation pairs for each voxel to obtain a single, within-network correlation value for that voxel, for each participant. We then calculated for each voxel in the social orienting network its correlation to each voxel in network 2, and averaged across all these correlation pairs to obtain a single across networks correlation value for that voxel for each participant. The relative strength of these within vs. across-network correlations is shown in Fig. 6a, b, where the within vs. across-network correlation strengths for each voxel were averaged across participants in either the matched TD group (Fig. 6a) or the ASD group (Fig. 6b). We next identified in the matched TD group the voxels whose correlation within network was stronger than their correlation across networks (those with a positive value in panel a). The distribution of these same voxels in the ASD group is shown in Fig. 6c, and these are also the voxels colored red in the scatter plot shown in Fig. 6d. For Fig. 6e–h, the same analysis was repeated, but with the roles of the social orienting network and network 2 reversed. For the estimation of *p*-values reported in the text which were used to determine whether there was a significant difference between within network and across-network correlations, we calculated for each participant the average within-network correlation for each of the two networks, and the average across-network correlation, and then carried out a paired two-tailed *t*-test across participants, on the within-network vs. across-network correlations. This was done separately for the matched TD group and the ASD group. This is a more conservative estimate than calculating the average within vs. across correlation values for each voxel (averaged across participants), and then carrying out a paired *t*-test on the within vs. across correlations values across voxels.

**Reporting summary**. Further information on research design is available in the Nature Research Reporting Summary linked to this article.

## Code availability

All code will be available upon request. Please contact the corresponding author.

## Data availability

Data are available through NIH Figshare (10.35092/yhjc.c.4741556). Unthresholded maps for the images depicted in Figs. 3–5, as well as Supplemntary Figs. 1 and 2 are available through Neurovault: https://identifiers.org/neurovault.collection:6090.

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

## Acknowledgements

We thank Adrian Gilmore, Andrew Persichetti, and Stephen Gotts for many helpful conversations and insights, and Kelsey Csumitta for help with recruitment. This work was supported by the Intramural Research Program, National Institute of Mental Health (ZIAMH002920), clinical trials number NCT01031407.

## Author contributions

M.R. and A.M. conceived and designed the study. M.R., C.W., and G.R. collected and analyzed the data. C.W. created the experimental stimuli. M.R. and A.M. wrote the paper and all authors commented on the paper.

## Competing interests

The authors declare no competing interests.
