## [Peer Review File · Communications Biology]

Reviewers' comments:

Reviewer #1 (Remarks to the Author):

In this manuscript titled "Distinct neural mechanisms of social orienting and mentalizing revealed by independent measures of neural and eye movement typicality", the authors address questions of typicality (intersubject correlation) of brain activity while viewing naturalistic movie stimuli and its relation to typicality (based on Euclidean distance) of eye movement patterns in individuals with autism spectrum disorder (ASD) and typically developed (TD) controls. They show that typicality of both measures differs between the groups, particularly showing higher variability in individuals with ASD, in support of prior findings. Additionally, in distinct brain regions, typicality of individuals' brain responses are related to the typicality of eye gaze patterns that, crucially, are derived from behavior while watching an independent social stimulus outside of the scanner. This independent measure of gaze behavior, its apparent stability in different subsets of videos (although see a specific comment on a suggested validation analysis) as well as connecting the two measures in both groups of individuals are significant strengths of the study compared to prior work assessing both measures using same stimuli separately. This work is a potentially important contribution to the field, but some additional analyses to further verify the stability and interpretation of the findings and a more thorough explanation of the analysis methods would be required. Additionally, some relatively minor revisions and additions to the text could strengthen the paper.

The comments are presented in the order they appear in the manuscript because all of the observations are relatively minor, assuming the additional analyses support the initial interpretations and the insufficiently explained data analyses prove to be appropriate.

1. Page 2. Second paragraph claims that "attentional synchrony ... is only very weakly modulated by higher level comprehension of the narrative, or inferences based on the mental states of the characters in the scene...".

This does not seem accurate. Attentional and brain synchrony depends on the amount of control asserted by the director (Hasson et al. 2008, *Projections* 2: 1-28), subsequent memory of events (Hasson et al. 2008, *Neuron* 57: 452-62; Chen et al. 2017, *Nat Neurosci* 20: 115-125), top-down tasks (Lahnakoski et al. 2014, *NeuroImage* 100: 316-324) and synchrony is also shown to be related to similarity of people such as friends vs. strangers (Parkinson et al. 2018, *Nat Comm* 9: 332). These are not "very weak" but in fact quite striking effects, in many cases. Thus, the authors should elaborate on the previous behavioral effects and what they mean exactly with the "weak" attentional effects.

2. Related to the first point, on Page 3, paragraph 2, it reads: "Eye movement patterns, reflecting attentional selection, vary across individuals, as do all aspects of their social comprehension, from basic understanding to empathy for the characters in the scene".

How is this compatible with the claim that attention is dominated by low-level visual changes and social content as claimed above?

3. End of page 3: "Independent measure of behavior ... allowed us to make inferences which generalize beyond our specific movie stimuli".

It is not immediately clear, until the end of the next sentence, what is meant by independent measure of behavior and why it generalizes. Stating immediately that the eye movements were recorded during (two different sets) of independent movie stimuli would avoid some potential confusion.

4. Page 4, first paragraph: "second, distinct network, ... does not correlate with the eye movements". It is not entirely clear that there is no correlation with eye movements. There is no correlation in the

TD group (Fig. 4A) but across both groups, there seems to be a correlation (Fig. 4B). This effect likely is driven by the group differences in both measures, but repeating the analysis also in the ASD group and, additionally, with partial correlations controlling for the mean group effects (that is, subtracting the mean ISCs and mean gaze typicality of each group and repeating the correlations) could clarify whether the correlation effect is driven only by the group or if there is a weaker gaze dependence when both groups are included in the analysis that may potentially be present only in the ASD group.

5. Figures. Some of the font sizes are not legible without zooming in considerably. Particularly in the legends for the line/dot colors in Figure 1 & 2 and, to a lesser degree, colorbar in Figure 4.

6. Page 7 paragraph 2 it is stated that real stability values for gaze typicality “were all entirely outside the random distribution”. It would be more informative to show this random distribution as a histogram with the true value (distributions) overlaid on top. This would naturally fit as a second panel in Figure 2.

7. Page 7, last paragraph: “but the variance in typicality of the ASD group was significantly higher than in the matched TD subset (533 and 163, respectively...”

Are these the variances of Euclidean distances measured in pixels? Measures in mm or degrees in visual angle would seem more customary.

8. Page 7, last paragraph: “This variance in behavior within the ASD group was not explained by social behavior”

This is a rather unclear statement. Maybe call it e.g. “social impairment” or “social traits” because gaze behavior while watching social interactions is also social behavior.

9. Page 10, paragraph 1 of section “Correlation between eye movement typicality and neural typicality” talks about the significance of anticorrelations between eye movement and brain activity typicality. What is the level of significance and how is it calculated? In the next paragraph only the permutation-based cluster correction threshold is mentioned.

10. Page 10, last paragraph: Why is the same analysis not done for the ASD group as was done with the TD group in the previous paragraph? The direct group comparison may be more difficult, but still it seems like an illogical choice to jump to averaging based on an ROI from TD results without first showing (qualitatively at least) whether the regional analysis results were similar in both.

11. Figure 4: The color scale in panel B seems less than ideal. The green areas are covering the shape of the separate ASD and TD results. An additive colormap, where you can see the overlap of all three regions would be better suited to show the shape of all three results (ASD, TD and full group). Additionally, presenting also the analyses (ASD and partial correlation analyses) suggested in point 4 would allow for more clear interpretation of the results in this figure.

12. Page 14, “Two different networks” section. To allow the reader to also appreciate how the networks separate, it would be helpful to plot the (mean) correlation matrices between the voxels in these networks. This might have to be done after spatial downsampling to reduce the matrix size, e.g. taking every other voxel or resampling the whole figure. Alternatively/additionally, the histograms of within and between-network correlations would help in seeing how the within-network and between-network correlations are distributed. Preferentially, each network should be presented separately in such a plot. The

13. End of page 14 it states “Despite this lack of differentiation for the within vs. across correlations within network 2 for the ASD participants, the subset of voxels showing significantly greater within

than across network correlations in the TD participants overlapped almost entirely with the subset of voxels in the ASD group which were similarly more correlated within rather than across network, for both networks (99% overlap for voxels in the social orienting network, and 97% overlap for voxels in network 2)".

How is it possible that there is no difference in within vs. between network correlations for a network for one group (but there is in the other group), but they still look identical between groups? Showing more of the data and explaining this more thoroughly could presumably help in explaining this apparent discrepancy. Currently, in this paragraph, there are so many serial correlations that it is difficult to follow what exactly is correlated at each step.

14. Figure 5: These results could potentially tie in better with Figure 4 if the suggested partial correlation is included.

15. Figure 6: The distinctness (lack of overlap) of the networks seems artificial in this figure since it is enforced by the exclusion of the voxels. The message of distinct (but overlapping) networks would seem to be better conveyed by Supplementary Figure 2, which does not artificially segregate the networks and includes the overlapping region that is also discussed later (as potential integration zone).

16. Page 16, line 2: "two networks shown above was differentially".

Please indicate the figure number rather than "above", as the figure might not be above this place in the text after copy-editing.

17. Page 16 lines 4-5: "social orienting network correlated to the SRS at $r = -0.15$, $p = 0.38$ ".
It did not correlate, significantly at least.

Page 17, end of first paragraph of Discussion: "our prediction was that differences in neural typicality between a TD group and an ASD group ... would reveal a separate mentalizing network, which would not overlap with the social orienting network"

Why should the networks not overlap? Many brain regions participate in multiple networks, potentially at different times, so this expectation seems puzzling. Particularly, here the networks were forced to not overlap, but practically, there were regions that both showed a mean group effect and a linear correlation with gaze behavior (Fig. S2) so this expectation was not fulfilled(?). Again, the partial correlations could possibly delineate these better to make sure some of the differences are not driven by the partly different groups in the analyses.

18. Page 17, beginning of paragraph 2: "Eye movements have been used extensively to study social orienting, though to our knowledge this is the first time that such a measure of typicality has been used in this way".

What do the authors mean by "in this way"? Highly related measures of typicality have in fact been used previously. For example in Hasson et al. 2009 (also cited in this paper), used x and y correlations to measure eye gaze similarity in autism, which is a very similar approach to the Euclidean distance here. With a somewhat different approach, Lahnakoski et al. (2014, mentioned in point 1) used similarity of gaze locations to predict similarity of brain activity, but over time, not individuals. However, what sets the current approach apart is the independent, reliable measure of eye movements in a different (social) stimulus environment and using that as a predictor across participants. Being more specific and highlighting this strength here would seem more appropriate than an ambiguous statement of novelty.

19. Page 17, paragraph 3 (stability of eye movement typicality): "The highly significant correlations of eye movement typicality across different movie clips (Figure 2) pointed to this being a robust measure of social orienting at the level of the individual. This gave us an independent behavioral measure, with

which to search for the neural correlates of social orienting.”

As an additional validation, it would be important to check that these stable patterns are not driven e.g. by a constant difference in the calibration. That is, if the mean fixation location for someone is different from the typical (grand mean), then the Euclidean distance will show a stable shift for that individual. A supplementary analysis excluding this possibility could lay these worries to rest. Simply removing the mean from the coordinates and recalculating the distances should achieve this.

20. Page 18, last paragraph: Monkey electrophysiology, as well as lesion studies in both monkeys and humans, have all pointed to an important role for STS in gaze perception and social orienting These human neuroimaging studies are limited however by the very simplistic and specific nature of their stimuli (mostly cartoon gaze cues), which is perhaps why we were able to uncover a broader network linked to social orienting...”

The comparison here seems somewhat arbitrary. The papers mentioned in this paragraph focus on perception of gaze cues, not active social viewing behavior of the individual being measured as is done in the current study. While it is reasonable to assume some similarity for brain substrates responsible for performing and observing social orienting, the analyses here do not seem to address observing social orienting in others and thus there is no reason to expect that the networks would be identical.

21. Page 19, toward the end of paragraph 2: “Amygdala on the other hand, is consistently found to be crucial for emotional processing”

Emotion processing may not be the only relevant thing here. For example, the amygdala also seems important for social/saliency perception, at least from faces (see e.g. Adolphs 2008 Curr Opin Neurobiol 18: 166–172). Thus, the amygdala should not be portrayed as being uniquely specialized for emotions as this type of reverse correlation may severely bias interpretation.

22. Page 19, last two lines: “together with the consistency of greater within compared to across network functional connectivity (Figure 6), strongly point to these being two distinct networks.”

Again, arguably, the analyses force the networks to be separate from each other. While it seems that there is indeed a subset of regions that show group differences that are related to the (stable) typicality of individuals, it may be too strong a claim that these are entirely distinct networks. As said in the next paragraph, there is in fact overlap when the networks are not forced to be non-overlapping. Additionally, for example, the eye movements during the same stimulus might still correlate with the brain ISC (state-like typicality) in regions that do not correlate with stable (trait-like) typicality, so they might not be entirely independent.

23. Page 21, Eye tracking setup: “Movies were viewed on a digital monitor with a 1920 x 1080 resolution”

Was this full screen? What was the screen size? More information on the visual stimulus system should be expected.

24. Page 22, first paragraph: “All movies were presented with audio.”

Again, more specifics about the stimulus delivery systems would be beneficial to allow others to replicate the methods. Not all audio and video systems are equal and may affect the responses that can be expected.

25. Page 22, title “Imagining data collection, MRI parameters and preprocessing”

Perhaps the authors mean “Imaging”?

26. Page 22, paragraph on post-hoc preprocessing states: “Volumes were slice-time corrected”

It was stated earlier that an online slice time correction was applied during data acquisition. Is this post-hoc correction then not going to misalign the slices again? This likely has a negligible effect on

the results with a 2-second TR, so it is not a major concern, but the authors should still consider verifying that they are not introducing unwanted timeshifts in the data.

27. Page 22, same paragraph: "whole signal from BOLD sources increases linearly over echo times, signals from non-BOLD sources remain constant across echoes."

This is not clear, what is the "whole signal from BOLD sources" and why do non-BOLD signals not change between the echos? Is this true for all non-BOLD sources?

28. Page 23, top: "Voxels with a temporal signal to noise ratio under 40 were removed from further analysis."

How was the tSNR estimated?

29. Page 23, Data analysis and statistical tests: "Voxels were considered significant if they were significant in any of the corrected cluster sizes for the corresponding threshold."

So, if I understand correctly, this is essentially making four partly dependent corrections with different initial threshold and taking the most liberal threshold? It sounds like this could inflate the false positive rate to some extent, but because of the dependence of the tests, this might not be a big problem. Were there multiple clusters that only survived one of these corrected size thresholds? Seeing the overlap of the four analyses would be crucial for understanding the stability of the results rather than showing everything that survives at least one test.

30. Data analysis section is extremely short. There are several analyses in the manuscript correlating various things where statistics are not entirely clear. Moreover, what how were the t-tests performed when comparing the within vs. between network correlations. That is, were the correlations first averaged across voxel pairs to get independent values for each subject or were all the voxel pairs used as independent measures? This section should be extended to sufficiently cover all relevant details.

31. Data availability: What data exactly will be made public. Is the code for repeating the analyses also included?

32. Supplementary figure 1 title: "Correlations between neural typicality and eye movement typicality using across group definitions"

What does "using across group definitions" mean?

Reviewer #2 (Remarks to the Author):

The research was aimed at studying brain regions responsible for social interactions. For this purpose, a group of people with typical development and with Autism Spectrum Disorder were engaged.

The studies were based on an experiment in which selected videos were used for exploring typicality of eye movement and neural responses. The collected data was studied, taking various aspects into account. The analysis included the assessment of the obtained results in terms of their correlations in regard to brain regions and eye movements.

However, the presentation of the eye-tracking data processing is too general. The method for selecting fixations and scan-paths comparison should be described better. Were scan-paths built from fixation or each registered sample? Please provide an example for calculating a distance between scan-paths.

The studies can be interesting for prospective readers and can be further developed. However, in some parts, the text is a little bit difficult to read. There are some long sentences, including additional information in brackets, which makes contents understanding demanding.

Reviewer #3 (Remarks to the Author):

Reviewers: Mikko Sams and Enrico Glerean

The authors studied the components of the social brain using neurotypical and individuals with autism. They measured eye movement and neural typicality, when participants were viewing Hollywood movie clips depicting social interactions. Brain activity was studied by inter-subject correlation, a measure of similarity and dissimilarity of brain activity of different individuals. Importantly, typicality of eye movements was studied with the different material as used in the fMRI experiment. Typicality of eye movements was compared with typicality of neural activity. Another approach to the data was direct comparison of group neural similarity of neurotypical individuals and individuals with autism. The results revealed two component networks of the social brain, "social orienting" and "inferring mental states of others".

I especially appreciate the eye movement part of this MS. I tend to think that the MS would be much clearer and useful for the readers, if it would contain only this part: typicality of eye movements and neural activity, and how they are related.

Abstract

This could be improved, it should be made clear how these two networks were found. Perhaps there is no enough space in the Abstract, but then in introduction it should be made clear if these networks were previously known, and now the authors introduce a new method for teasing them apart.

Introduction

Give a specific and clear definition to Social orienting network.

Results

As mentioned by the authors, one limitation of the study is that quite important some top and bottom parts of the brain could not be scanned. I take this study more as a methodological one, so in my opinion this is not a crucial limitation. Anyhow, the authors should lengthen their discussion on this issue. What have others found in these areas, which could be relevant for this study? ISC is not an acronym for average neural typicality.

Mark the anatomical areas you are talking about in the pictures, it is important to make reading of the figures easy.

Show neural typicality maps also for the ASD group (cf. Figure 3)

Do I now understand this correctly? The "social orienting network" is the one, which can be found both in TD and ASD participants, and which covaries with the typicality of eye movement patterns. In some parts of this network, neural similarity of TD and ADS participants differ. The second network is found by directly combining neural typicality of TD and ASD subject, involving those areas where there. This is what has been done before in previous studies, so this analysis should confirm and perhaps extend

results obtained in them.

The part of the MS analyzing the differences of these two networks is not very easy to read. Perhaps it would be better to start with the comparison of neural similarity between TD and ASD participants, as done previously. And then going to the analysis using eye movements, as a new behavioural method, which then shows something new, a social orienting network. This network then appears to be quite similar in TD and ASD participants, even though within it neural similarity is smaller in ASD participants. But there is additional difference found in comparing neural similarity, which is not correlating with eye movements. This then is suggested to be related to mentalizing.

I have difficulties in following the reasoning here. It may be correct, but should be written more clearly for a basic human neuroscientist to follow the story and be convinced of the reasoning. So my suggestion is to spend more time on this, also being more economical in writing.

Methods

Give the details of the film moving during eye tracking in enough details that the experiment could be replicated.

Please give the IQs and other this type of information of the subjects.

It is not clear how the intersubject correlation was computed: was it done pairwise (as recommended by previous works of Gang Chen or Jussi Tohka), or is it that the time series of a subject is correlated to the average of all other N-1 subjects? From the manuscript it seems that the time series of a subject is correlated with the average across all subjects (including the subject of interest), if this is the case then the statistics will be biased as there is always a non zero correlation. Opening the code (see below) would increase the transparency of the methods. Similar concerns apply to the measure of typicality for eye tracking: was it a correlation between an individual time series with the average of all participant' time series (including the subject of interest)? These procedures should be clarified in the methods section.

Correlation between eye movement typicality and neural typicality: is this done across pairs (i.e. mantel test) so that the correlation value of a subject pair brain time series is compared to a correlation value of their eye tracking time series? Or is this done across individual subjects? Please clarify in the methods section.

It is not clear if the p and q values used in figure 5 were derived from the t-statistics or if they were computed with permutation-based clustering. Since in the methods you specify that all correction for multiple comparisons were done with permutation-based clustering, the same approach should also be used in figure 5.

Please upload all unthresholded statistical brain maps to NeuroVault and include the neurovault link into the manuscript. NeuroVault is a tool developed at Stanford to share result maps, this enables future meta-analyses as well as allowing the readers to explore the results interactively. Neurovault collection is then automatically linked to the paper after the paper is published, increasing its visibility.

Code: you mentioned in-house software written in MATLAB. While this is not mandatory, for transparency it is highly recommended to share the code used. Even if the code is not perfect, or optimised, or documented, it is still better than having no code at all.

It seems that data can be shared, which is fantastic. Please include already the link to the data

repository in the manuscript to make the data FAIR.

Perhaps it would be more sensitive to use non-Hollywood clips to have more natural differences in gaze patterns? Please discuss.

Reviewer #1 (Remarks to the Author):

In this manuscript titled "Distinct neural mechanisms of social orienting and mentalizing revealed by independent measures of neural and eye movement typicality", the authors address questions of typicality (intersubject correlation) of brain activity while viewing naturalistic movie stimuli and its relation to typicality (based on Euclidean distance) of eye movement patterns in individuals with autism spectrum disorder (ASD) and typically developed (TD) controls. They show that typicality of both measures differs between the groups, particularly showing higher variability in individuals with ASD, in support of prior findings. Additionally, in distinct brain regions, typicality of individuals' brain responses are related to the typicality of eye gaze patterns that, crucially, are derived from behavior while watching an independent social stimulus outside of the scanner. This independent measure of gaze behavior, its apparent stability in different subsets of videos (although see a specific comment on a suggested validation analysis) as well as connecting the two measures in both groups of individuals are significant strengths of the study compared to prior work assessing both measures using same stimuli separately. This work is a potentially important contribution to the field, but some additional analyses to further verify the stability and interpretation of the findings and a more thorough explanation of the analysis methods would be required. Additionally, some relatively minor revisions and additions to the text could strengthen the paper.

The comments are presented in the order they appear in the manuscript because all of the observations are relatively minor, assuming the additional analyses support the initial interpretations and the insufficiently explained data analyses prove to be appropriate.

We would like to thank the reviewer for this extremely thorough and thoughtful review. We are grateful for the reviewer's appreciation for the merits of our work, and we do not take for granted the degree of care and consideration which were invested in this review. We have attempted to address all of the reviewer's concerns with the same care and thoughtfulness.

1. Page 2. Second paragraph claims that "attentional synchrony ... is only very weakly modulated by higher level comprehension of the narrative, or inferences based on the mental states of the characters in the scene...".

This does not seem accurate. Attentional and brain synchrony depends on the amount of control asserted by the director (Hasson et al. 2008, *Projections* 2: 1-28), subsequent memory of events (Hasson et al. 2008, *Neuron* 57: 452-62; Chen et al. 2017, *Nat Neurosci* 20: 115-125), top-down tasks (Lahnakoski et al. 2014, *NeuroImage* 100: 316-324) and synchrony is also shown to be related to similarity of people such as friends vs. strangers (Parkinson et al. 2018, *Nat Comm* 9: 332). These are not "very weak" but in fact quite striking effects, in many cases. Thus, the authors should elaborate on the previous behavioral effects and what they mean exactly with the "weak" attentional effects.

We thank the reviewer for pointing out that the distinction between attentional synchrony and neural synchrony was not made sufficiently clear in the introduction. The evidence points to attentional synchrony (as measured by eye movements) being driven primarily by lower-level visual and social cues (such as gaze direction, body language). Neural synchrony on the other hand, appears to capture much higher-order cognitive processes, consistent with a shared experience of the movie at these higher levels. We have expanded the introduction to better elaborate on this distinction (pp. 1-2):

Directors of Hollywood movies are particularly adept at manipulating the focus of our attention, using cinematic techniques to tightly control where viewers' attention is drawn¹⁰⁻¹³. However, attentional synchrony, even when anchored around social cues, need not be driven by higher order cognition or mentalizing, as argued convincingly regarding the non-human primate literature¹⁴. Similarly, in humans, attentional synchrony seems to be dominated by transient visual and social cues, and is only very weakly modulated by higher level comprehension of the narrative, or inferences based on the mental states of the characters in the scene, as is demonstrated by studies which manipulated comprehension through temporal shuffling of scenes¹⁵, or manipulation of available context^{16,17}.

However, movie experiences are generally robustly shared across viewers at higher cognitive levels as well. Previous studies have described widespread correlations in neural responses across individuals, extending well beyond perceptual regions into higher level processing regions¹⁸. This neural synchrony, or neural typicality, measured by the inter-subject correlations (ISC) of the neural response time course to the movie, has been shown to underlie shared subsequent memories for events in the movie¹⁹, a shared interpretation of the narrative²⁰, and can even predict friendship²¹. Thus there appears to be a distinction between the lower-order process of choosing what to attend to, and higher-order processes involved in the complex interpretation of what we saw. In the framework of movie viewing, similarity in eye movement patterns reflects similarity in mechanisms of social orientation, whereas similarity in social comprehension and mentalizing would only be reflected in the degree of similarity of the neural responses in the brain regions involved in these tasks, and not in eye movement patterns.

2. Related to the first point, on Page 3, paragraph 2, it reads: "Eye movement patterns, reflecting attentional selection, vary across individuals, as do all aspects of their social comprehension, from basic understanding to empathy for the characters in the scene". How is this compatible with the claim that attention is dominated by low-level visual changes and social content as claimed above?

We hope this point has been clarified by the above addition to the introduction. Social orienting and social comprehension are two separate functions, and there is individual variance in both. While variability in social orienting will be reflected in the variance of the eye movements (as well as in the neural responses in relevant regions), variability in social comprehension will be reflected only in the variance of the neural responses in the regions involved in those higher order tasks.

3. End of page 3: “Independent measure of behavior ... allowed us to make inferences which generalize beyond our specific movie stimuli”.

It is not immediately clear, until the end of the next sentence, what is meant by independent measure of behavior and why it generalizes. Stating immediately that the eye movements were recorded during (two different sets) of independent movie stimuli would avoid some potential confusion.

To better stress this point, we have added the sentence in red to the text to the first sentence (p.4):

*Here we sought to exploit the full capacity of the movie viewing environment by expanding the analysis to include an independent measure of behavior, **with eye movements recorded during a different, independent set of movie clips than that used for the fMRI session.** This allowed us to make inferences which generalize beyond our specific movie stimuli.*

4. Page 4, first paragraph: “second, distinct network, ... does not correlate with the eye movements”. It is not entirely clear that there is no correlation with eye movements. There is no correlation in the TD group (Fig. 4A) but across both groups, there seems to be a correlation (Fig. 4B). This effect likely is driven by the group differences in both measures, but repeating the analysis also in the ASD group and, additionally, with partial correlations controlling for the mean group effects (that is, subtracting the mean ISCs and mean gaze typicality of each group and repeating the correlations) could clarify whether the correlation effect is driven only by the group or if there is a weaker gaze dependence when both groups are included in the analysis that may potentially be present only in the ASD group.

This is an important point, and we now address this in supplementary figure 1. See also response to points 10 and 11 below.

5. Figures. Some of the font sizes are not legible without zooming in considerably. Particularly in the legends for the line/dot colors in Figure 1 & 2 and, to a lesser degree, colorbar in Figure 4.

We thank the reviewer for pointing out this difficulty in legibility. We have enlarged the fonts for these figures.

6. Page 7 paragraph 2 it is stated that real stability values for gaze typicality “were all entirely outside the random distribution”. It would be more informative to show this random distribution as a histogram with the true value (distributions) overlaid on top. This would naturally fit as a second panel in Figure 2.

We have added this analysis as a second panel to Figure 2

2a

b

7. Page 7, last paragraph: “but the variance in typicality of the ASD group was significantly higher than in the matched TD subset (533 and 163, respectively...)” Are these the variances of Euclidean distances measured in pixels? Measures in mm or degrees in visual angle would seem more customary.

These are indeed pixels, but essentially they are arbitrary units. Since we are averaging across all the frames across the entire movie set, it is not clear what meaning could be inferred from mm or degrees of visual angle.

8. Page 7, last paragraph: “This variance in behavior within the ASD group was not explained by social behavior” This is a rather unclear statement. Maybe call it e.g. “social impairment” or “social traits” because gaze behavior while watching social interactions is also social behavior.

Social impairment is indeed a more accurate term, we have corrected this in the text.

9. Page 10, paragraph 1 of section “Correlation between eye movement typicality and neural typicality” talks about the significance of anticorrelations between eye movement and brain activity typicality. What is the level of significance and how is it calculated? In the next paragraph only the permutation-based cluster correction threshold is mentioned.

We thank the reviewer for pointing out this omission. Significance in this case was determined by calculating the p-value of the correlation coefficient, given the sample size, at a threshold of $p < 0.05$. We have added this information to the text (p.11).

10. Page 10, last paragraph: Why is the same analysis not done for the ASD group as was done with the TD group in the previous paragraph? The direct group comparison may be more difficult, but still it seems like an illogical choice to jump to averaging based on an ROI from TD results without first showing (qualitatively at least) whether the regional analysis results were similar in both.

We thank the reviewer for this comment. We had previously decided to omit this analysis from the manuscript, as the direct group comparison is indeed difficult to carry out. A qualitative comparison however does seem to be in order. We have added the results of the same analysis for the ASD as the first panel in supplementary figure 1:

11. Figure 4: The color scale in panel B seems less than ideal. The green areas are covering the shape of the separate ASD and TD results. An additive colormap, where you can see the overlap of all three regions would be better suited to show the shape of all three results (ASD, TD and full group). Additionally, presenting also the analyses (ASD and partial correlation analyses) suggested in point 4 would allow for more clear interpretation of the results in this figure.

Following these last two comments, and comment 4 above, the new supplementary figure now shows both the analysis for the ASD group alone (supplementary figure 1a), and the analysis for the joint group after removing mean ISC and mean gaze typicality, suggested in comment 4 (supplementary figure 1b). We have also made the following changes to the text to reflect this (pp. 11-12):

The results of the same analysis correlating eye movement typicality with neural typicality for the ASD group are displayed in Supplementary Figure 1a, and are centered on very similar regions.

To directly test whether the same network which we found in TD participants also underlies social orienting in participants with ASD, we examined the correlation between eye movement typicality and neural typicality within the network defined by the TD group for the ASD group.

...

To ensure that these correlations across the combined TD and ASD group were not driven purely by group differences in eye movement typicality and/or neural typicality (see below), we carried out an additional analysis controlling for the mean group effects. In this analysis, we subtracted the group average for both eye movement typicality and neural typicality from each of the TD and ASD groups, before combining the two. The results are displayed in Supplementary Figure 1b. When the variance accounted for by the group means is removed, the social orienting network revealed by this analysis is more limited than that seen in Figure 4b, and more similar to the results for the separate TD and ASD groups (Figure 4a, Supplementary Figure 1a).

12. Page 14, “Two different networks” section. To allow the reader to also appreciate how the networks separate, it would be helpful to plot the (mean) correlation matrices between the voxels in these networks. This might have to be done after spatial downsampling to reduce the matrix size, e.g. taking every other voxel or resampling the whole figure. Alternatively/additionally, the histograms of within and between-network correlations would help in seeing how the within-network and between-network correlations are distributed. Preferentially, each network should be presented separately in such a plot. The

We thank the reviewer for highlighting this explanatory gap in the manuscript. We have added this additional figure to the paper (new Figure 6). We have also added a reference to the specific panels each statistic refers to in the text.

Figure 6: within network correlations vs. across network correlations. Distribution of within network correlations vs. across network correlations for all the non-overlapping voxels identified in the analyses portrayed in Figure 4 (social orienting network) and Figure 5 (network 2). (a) Histograms shows the average correlation of each voxel in the social orienting network to all other voxels in the social orienting network, minus its average correlation to all voxels in network 2, averaged across all matched TD participants. Vertical black line at 0 denotes equal correlation, i.e. voxels with this value are equally correlated to the other voxels in the social orienting network and to voxels in network 2. Voxels to the left of this line are more correlated across networks than within network (i.e. higher correlation to voxels in network 2), and voxels to the right are more correlated within network than across networks. (b) Same for the ASD group. Note the significant rightward shift in both these plots, which is reflected in the statistical difference cited in the text. (c) Distribution in the ASD group of all the voxels in the social orienting network which were more correlated within the social orienting network than to the voxels in network 2 in the TD group, using the TD indices for these voxels. 99% of voxels which were more correlated within the social network

than across networks in the TD group were also more correlated within network in the ASD group. (d) Scatter plot of the same analysis shown in panels b-c. Each dot represents one voxel in the social orienting network, with its value on the x-axis reflecting its average correlation to all the other voxels in the social orienting network, and value on the y-axis reflecting its average correlation to all voxels in network 2, averaged across all ASD participants. Identity line (i.e. equal correlation to both networks) marked in black. Blue dots are voxels which were less correlated within than across network in the matched TD group, whereas red dots are the voxels which had greater within network than across network correlations in the matched TD group. Note the almost complete correspondence of these voxels across the two populations. Panels e-h show the same analyses as above, for the correlations of voxels in network 2 to all other voxels in network 2 vs. their correlations to the voxels in the social orienting network.

13. End of page 14 it states “Despite this lack of differentiation for the within vs. across correlations within network 2 for the ASD participants, the subset of voxels showing significantly greater within than across network correlations in the TD participants overlapped almost entirely with the subset of voxels in the ASD group which were similarly more correlated within rather than across network, for both networks (99% overlap for voxels in the social orienting network, and 97% overlap for voxels in network 2)”.

How is it possible that there is no difference in within vs. between network correlations for a network for one group (but there is in the other group), but they still look identical between groups? Showing more of the data and explaining this more thoroughly could presumably help in explaining this apparent discrepancy. Currently, in this paragraph, there are so many serial correlations that it is difficult to follow what exactly is correlated at each step.

We thank the reviewer for pointing out this issue. In creating the new Figure 6 we have gone back and re-analyzed these data and reran the statistical tests. We found a mistake in the original code, which impacted these statistics. While the p-values for most of these tests did not change in a way that changes the meaning of the results, the p-value for this one test did change considerably. Here is the paragraph with the amended statistics, with the original statistics added in red (p.17):

*There was a significant difference for the social orienting network, with correlations within the social orienting network significantly greater than correlations between the social orienting network and network 2 for both the TD and the ASD groups (Figure 6a-b, paired two-tail t-test, $p = 5.4 \times 10^{-12}$ and $p = 5.5 \times 10^{-13}$ accordingly [**was $p < 7.4 \times 10^{-14}$ for both groups**]). Correlations within network 2 were also significantly more correlated within than across networks for both TD and ASD participants (Figure 6e-f, $p = 4.4 \times 10^{-8}$ for TD participants ([**was $p = 2.9 \times 10^{-8}$**]), and $p = 0.036$ for ASD participants). Importantly, not only was there an overall significant bias for within vs. across network correlations, but the subset of voxels showing significantly greater within than across network correlations in the TD participants overlapped almost entirely with the subset of voxels in the ASD group which were similarly more correlated within rather than across network, for both networks (99% overlap for voxels in the social orienting network, Figure 6c-d, and 94% ([**was 97%**]) overlap for voxels in network 2, Figure 6g-h).*

We apologize for the confusion this has caused, and have gone back to check the code for all the other statistical tests in this paper, but have found no further problems. As we say also in response to comment 31, we will make all of the code available upon request. We expand upon this point in response to comment 30 about the data analysis section in the methods, where we explain that we in fact chose the most conservative statistical test, testing across participants rather than across voxels. If we test across voxels instead, the p value for the ASD group in within network 2 correlation to across network correlation is 6.4×10^{-11} , and the other three test have p-values below 1.5×10^{-32} . We also hope that these results are now more readily understandable with the addition of the new Figure 6.

14. Figure 5: These results could potentially tie in better with Figure 4 if the suggested partial correlation is included.

This is indeed the case, and the suggested partial correlation is important for understanding the data. However, it also artificially constrains the social orienting network from including regions with significant neural typicality differences between the groups, as these are removed prior to the analysis. Therefore, this analysis cannot find any overlap between the analyses in Figure 4 and 5, even if those exist.

15. Figure 6: The distinctness (lack of overlap) of the networks seems artificial in this figure since it is enforced by the exclusion of the voxels. The message of distinct (but overlapping) networks would seem to be better conveyed by Supplementary Figure 2, which does not artificially segregate the networks and includes the overlapping region that is also discussed later (as potential integration zone).

As can be seen in the histograms, the exclusion of voxels did not in fact force the segregation of the networks as we have defined them. The distinctness of the networks displayed in Figure 7 (previously Figure 6) is defined by the voxels within those networks being more correlated within network than across network, which is quite a standard network definition in resting state fMRI. New Figure 6a,b,e,f shows that simply excluding the overlapping voxels from the previous analyses does not ensure this connectivity profile, which is not surprising considering the other analyses were not defined by connectivity (between voxels) in any way. However, the striking result from Figure 6c,d,g,h is how stable these networks are across populations, with regard to the identity of the subset of voxels. The subset of the voxels within each of these networks (using the definitions in the previous analyses) which has the desired connectivity pattern of greater within than across correlations in the TD group, overlaps almost entirely with the subset of voxels showing this connectivity pattern in the ASD group, lending credence to the notion that these are in fact functionally distinct networks. We have also added a clarification to the legend of Figure 7,

16. Page 16, line 2: “two networks shown above was differentially”. Please indicate the figure number rather than “above”, as the figure might not be above this place in the text after copy-editing.

Done.

17. Page 16 lines 4-5: “social orienting network correlated to the SRS at $r = -0.15$, $p = 0.38$ ”.

It did not correlate, significantly at least.

Changed the wording in the text to read (p.19):

whereas the neural typicality averaged across the social orienting network was not significantly correlated to the SRS at $r = -0.15$, $p = 0.38$

Page 17, end of first paragraph of Discussion: “our prediction was that differences in neural typicality between a TD group and an ASD group ... would reveal a separate mentalizing network, which would not overlap with the social orienting network”
Why should the networks not overlap? Many brain regions participate in multiple networks, potentially at different times, so this expectation seems puzzling. Particularly, here the networks were forced to not overlap, but practically, there were regions that both showed a mean group effect and a linear correlation with gaze behavior (Fig. S2) so this expectation was not fulfilled(?). Again, the partial correlations could possibly delineate these better to make sure some of the differences are not driven by the partly different groups in the analyses.

Indeed, this point was not stated clearly enough in the manuscript. Our expectation was that these would not be the same network, for reasons that are now hopefully better explained in the introduction (see response to points 1-2). It is of course possible that the mentalizing network would partially overlap with the social orienting network, as in fact appears to be the case from our data, and yet certain elements of this network should be distinct, as these two behaviors (social orientation and mentalizing) can diverge. We have rewritten the beginning of the discussion, to better frame our results, and have clarified the reasoning for our expectation that the networks be at least partially though not necessarily wholly divergent (new text in red, pp. 19-20):

Movie viewing evokes both shared neural responses, and shared behavior, in the form of eye movements orchestrated by carefully constructed visual, auditory and social cues. Yet there is individual variation in both behavior and neural responses. Evidence from previous studies suggests that neural typicality, or correlations between subjects in their neural responses, indicate shared processing, or a shared experience relevant to the function of the correlated region. For example, studies have demonstrated that for participants viewing a movie, or listening to a narrative, the similarity of their interpretation of the story predicts the degree of neural similarity in regions involved in narrative interpretation, such as the fronto-parietal network and the default mode network²⁰. Likewise, participants recalling the same events have been shown to have more neural similarity than participants recalling different events¹⁹. ASD participants have difficulties in both social orienting, and higher order mentalizing. These are likely the result of different processes, as previous studies have demonstrated a decoupling of high order comprehension / mentalizing during movie viewing from eye movement patterns¹⁵.

Using our measures of the correlation between eye movement typicality and neural typicality, and group differences between the TD and ASD groups in neural typicality, we therefore expected to find two different, partially overlapping networks. The correlation between the two typicality measures across participants was used to search for the network responsible for social orienting. As the neural typicality measure is an indicator of shared processing, we expected lower neural typicality in the ASD group in regions involved in the processes in which ASD participants differ from their TD counterparts, namely social orienting and higher order social comprehension / mentalizing. Therefore, we expected to find group differences in neural typicality in the regions underlying these behaviors.

For the social orienting network, on top of previous research²⁷⁻²⁹, our own data described here shows reduced typicality of eye movements in the ASD group compared with the TD group (Figure 2), and this should be reflected in the typicality of neural processing. However, at the single voxel level, we did not find group differences in neural typicality in most voxels of the social orienting network. Those differences were present at the network level though, as there was significantly lower neural typicality in the ASD group within the social orienting network as a whole, perhaps indicating a lack of sensitivity of the neural typicality group difference measure. The more robust effects, which could be seen at the single voxel level (Figure 5) centered on regions which seemed to constitute a separate, distinct network, whose correspondence to the mentalizing network will be discussed below.

18. Page 17, beginning of paragraph 2: “Eye movements have been used extensively to study social orienting, though to our knowledge this is the first time that such a measure of typicality has been used in this way”.

What do the authors mean by “in this way”? Highly related measures of typicality have in fact been used previously. For example in Hasson et al. 2009 (also cited in this paper), used x and y correlations to measure eye gaze similarity in autism, which is a very similar approach to the Euclidean distance here. With a somewhat different approach, Lahnakoski et al. (2014, mentioned in point 1) used similarity of gaze locations to predict similarity of brain activity, but over time, not individuals. However, what sets the current approach apart is the independent, reliable measure of eye movements in a different (social) stimulus environment and using that as a predictor across participants. Being more specific and highlighting this strength here would seem more appropriate than an ambiguous statement of novelty.

We thank the reviewer for pointing out the ambiguity in the original statement, and for suggesting a better framing for this statement. We have modified to text as suggested (p. 20):

Eye movements have been used extensively to study social orienting^{27,42,43}, though to our knowledge this is the first time that an independent measure of typicality, using a different set of stimuli and based on the stability of eye movement typicality as a measurable individual trait, has been used to predict similarity of neural activity.

19. Page 17, paragraph 3 (stability of eye movement typicality): “The highly significant correlations of eye movement typicality across different movie clips (Figure 2) pointed to this being a robust measure of social orienting at the level of the individual. This gave us an independent behavioral measure, with which to search for the neural correlates of social orienting.”

As an additional validation, it would be important to check that these stable patterns are not driven e.g. by a constant difference in the calibration. That is, if the mean fixation location for someone is different from the typical (grand mean), then the Euclidean distance will show a stable shift for that individual. A supplementary analysis excluding this possibility could lay these worries to rest. Simply removing the mean from the coordinates and recalculating the distances should achieve this.

We thank the reviewer for suggesting this important sanity check. We have added the results of this analysis to (p. 8):

To rule out the possibility that this stability in the typicality measure was a spurious result of a consistent calibration shift at the individual level, we repeated this analysis after demeaning the data per participant, and recalculated all the distances. The unshuffled mean correlations across iterations were very similar to before: $r = 0.71$ for all TD participants, $r = 0.63$ for the matched subset, and $r = 0.8$ for the ASD participants.

20. Page 18, last paragraph: Monkey electrophysiology, as well as lesion studies in both monkeys and humans, have all pointed to an important role for STS in gaze perception and social orienting These human neuroimaging studies are limited however by the very simplistic and specific nature of their stimuli (mostly cartoon gaze cues), which is perhaps why we were able to uncover a broader network linked to social orienting...” The comparison here seems somewhat arbitrary. The papers mentioned in this paragraph focus on perception of gaze cues, not active social viewing behavior of the individual being measured as is done in the current study. While it is reasonable to assume some similarity for brain substrates responsible for performing and observing social orienting, the analyses here do not seem to address observing social orienting in others and thus there is no reason to expect that the networks would be identical.

Indeed we neither expect nor find the networks to be identical, but rather the networks identified in these studies using very simplified stimuli appear to be a subset of the regions we identify using our more complex, ecological stimuli.

21. Page 19, toward the end of paragraph 2: “Amygdala on the other hand, is consistently found to be crucial for emotional processing”

Emotion processing may not be the only relevant thing here. For example, the amygdala also seems important for social/saliency perception, at least from faces (see e.g. Adolphs 2008 Curr Opin Neurobiol 18: 166–172). Thus, the amygdala should not be portrayed as being uniquely specialized for emotions as this type of reverse correlation may severely bias interpretation.

That is certainly a valid point. We have extended and qualified this statement (p. 23):

*Amygdala on the other hand, is consistently found to be crucial for emotional processing, as well as for social/saliency perception, among its many roles*⁵⁷⁻⁵⁹

22. Page 19, last two lines: “together with the consistency of greater within compared to across network functional connectivity (Figure 6), strongly point to these being two distinct networks.”

Again, arguably, the analyses force the networks to be separate from each other. While it seems that there is indeed a subset of regions that show group differences that are related to the (stable) typicality of individuals, it may be too strong a claim that these are entirely distinct networks. As said in the next paragraph, there is in fact overlap when the networks are not forced to be non-overlapping. Additionally, for example, the eye movements during the same stimulus might still correlate with the brain ISC (state-like typicality) in regions that do not correlate with stable (trait-like) typicality, so they might not be entirely independent.

As in response to point 15 above, our argument for distinctness relates only to the sub networks which are composed of the voxels which show greater within network than across network correlations, in a manner that is stable across both of our independent populations (which as can be seen from the data in the new Figure 6, virtually all these voxels are stable across populations). The more broadly defined networks, not limited by this connectivity analysis, indeed show also substantial overlap, as might be expected in such complex, high level cognitive tasks. We have clarified this distinction in the text (additions in red, p.23):

*The double disassociation between the two networks identified through the eye movement and neural typicality group difference analyses and our two behavioral measures (eye movement typicality, SRS), together with the consistency of greater within compared to across network functional connectivity (Figure 6), strongly point to **the two sub networks, which also show this pattern of connectivity**, being two distinct networks (Figure 7). Considering the nature of the processing that takes place while watching social movies, the nature of the deficits in ASD, and the previous literature on the regions identified by this analysis, we hypothesize that this second network revealed by the neural typicality group difference analysis corresponds to the mentalizing network. We also found substantial overlap between the two **networks using the broader definitions without the requirement of greater within than across network connectivity**, most notably in right IFG, MPFC and PCC, and bilaterally in regions of putamen and the caudate (Supplementary Figure 4).*

23. Page 21, Eye tracking setup: “Movies were viewed on a digital monitor with a 1920 x 1080 resolution”

Was this full screen? What was the screen size? More information on the visual stimulus system should be expected.

Added to the text (p. 24):

Movies were viewed full screen on a digital monitor with a 1920 x 1080 resolution, with a screen size of 20.5 x 12 inches.

24. Page 22, first paragraph: "All movies were presented with audio."

Again, more specifics about the stimulus delivery systems would be beneficial to allow others to replicate the methods. Not all audio and video systems are equal and may affect the responses that can be expected.

Added to the text (p. 25):

All movies were presented with audio, using the built in speakers on the MacBook Pro for the movie clips shown during the eye tracking portion of the experiment, and using the SilentScan headphone system from Avotec Inc. during fMRI scanning.

25. Page 22, title "Imaging data collection, MRI parameters and preprocessing"
Perhaps the authors mean "Imaging"?

Indeed. We have corrected this in the text.

26. Page 22, paragraph on post-hoc preprocessing states: "Volumes were slice-time corrected"

It was stated earlier that an online slice time correction was applied during data acquisition. Is this post-hoc correction then not going to misalign the slices again? This likely has a negligible effect on the results with a 2-second TR, so it is not a major concern, but the authors should still consider verifying that they are not introducing unwanted timeshifts in the data.

We have corrected this in the text. The source of the confusion was that there are two (identical) processing streams, one which uses the real-time volume registration and slice time correction, and another that starts with the raw dicoms, and then does offline volume registration and slice time correction. We now describe only the processing stream using the offline data.

27. Page 22, same paragraph: "whole signal from BOLD sources increases linearly over echo times, signals from non-BOLD sources remain constant across echoes."

This is not clear, what is the "whole signal from BOLD sources" and why do non-BOLD signals not change between the echos? Is this true for all non-BOLD sources?

This was a typo. We have corrected the paragraph to read (p.26):

*Briefly, this procedure utilizes the physical properties of BOLD and non-BOLD fluctuations, namely the fact that **while** signal from BOLD sources increases linearly over echo times, signals from non-BOLD sources remain constant across echoes.*

The multi-echo denoising procedure is described in depth in the cited reference, Kundu, P. *et al.* Integrated strategy for improving functional connectivity mapping using multiecho fMRI. *Proceedings of the National Academy of Sciences of the United States of America* **110**, 16187-16192. There is some debate about whether this method entirely removes non-BOLD sources of noise such as respiration, but this method has shown excellent results in denoising data, especially compared with other methods

28. Page 23, top: "Voxels with a temporal signal to noise ratio under 40 were removed from further analysis."

How was the tSNR estimated?

tSNR was estimated by dividing the mean of the signal for each voxel by the temporal standard deviation, averaged across participants. We have added this description to the text (p. 26).

29. Page 23, Data analysis and statistical tests: "Voxels were considered significant if they were significant in any of the corrected cluster sizes for the corresponding threshold."

So, if I understand correctly, this is essentially making four partly dependent corrections with different initial threshold and taking the most liberal threshold? It sounds like this could inflate the false positive rate to some extent, but because of the dependence of the tests, this might not be a big problem. Were there multiple clusters that only survived one of these corrected size thresholds? Seeing the overlap of the four analyses would be crucial for understanding the stability of the results rather than showing everything that survives at least one test.

This analysis takes into account that there are could be small but highly significant clusters. These might not be significant at the $p < .05$ threshold, for which the permutation testing might find large random clusters which are significant at this threshold. However, as significance levels increase, the size of contiguous random clusters decreases, so the cluster threshold at $p < .001$ is much smaller than the cluster threshold at $p < .05$. We therefore include both large clusters that are significant at lower significance thresholds, as well as smaller clusters that are significant at higher thresholds. This approach gives a less arbitrary and more complete, though no less stringent view of the data. We have added this description to the text (p. 27):

*Voxels were considered significant if they were significant in any of the corrected cluster sizes for the corresponding threshold, thus allowing the inclusion of both large clusters significant at lower thresholds, and smaller, highly significant clusters. This approach has been advised following the recent debate on cluster size corrections*⁶³.

30. Data analysis section is extremely short. There are several analyses in the manuscript correlating various things where statistics are not entirely clear. Moreover, what how were the t-tests performed when comparing the within vs. between network correlations. That is, were the correlations first averaged across voxel pairs to get

independent values for each subject or were all the voxel pairs used as independent measures? This section should be extended to sufficiently cover all relevant details.

We have substantially extended the data analysis section, so it now includes details for all the analyses in the manuscript. New text is marked in red (pp. 26-27):

All data were analyzed with in-house software written in MATLAB, as well as the AFNI software package. Data on the cortical surface were visualized with SUMA (Surface Mapping⁶²). Eye movement typicality for each of the TD participants was calculated per movie per frame, by computing the Euclidean distance of their average recorded eye position per frame to the mean recorded eye position of all other participants (i.e. all N-1 participants, not including the participant of interest) for each frame. These distances were then averaged for each participant across all frames of all movies, to obtain one typicality measure. Neural typicality was similarly assessed per participant, per voxel, by calculating the correlation of the time course of each voxel for that participant to the average time course of that voxel for all the other (N-1) participants. The correlation between the eye movement typicality and the neural typicality was calculated based on these two measures.

Two-tail t tests were used for all p values on correlations, unless otherwise stated. For the maps in Figure 4 which were corrected through a permutation-based cluster size correction, we permuted the subject labels for the eye tracking typicality for 10,000 iterations, and correlated these with the neural typicality. The cluster threshold was defined for each threshold separately ($p < 0.05$, $p < 0.01$, $p < 0.005$, $p < 0.001$) as the largest cluster at that threshold at the 95th percentile across all iterations. Voxels were considered significant if they were significant in any of the corrected cluster sizes for the corresponding threshold, thus allowing the inclusion of both large clusters significant at lower thresholds, and smaller, highly significant clusters. This approach has been advised following the recent debate on cluster size corrections⁶³. The same analysis was carried out for the results in Figure 5, permuting the TD and ASD labels for 10,000 iterations. For Figure 6a-d, we first calculated for each participant the correlations of each voxel within the social orienting network to every other voxel within the social orienting network, and averaged across all the correlation pairs for each voxel to obtain a single, within network correlation value for that voxel, for each participant. We then calculated for each voxel in the social orienting network its correlation to each voxel in network 2, and averaged across all these correlation pairs to obtain a single across networks correlation value for that voxel for each participant. The relative strength of these within vs. across network correlations is shown in Figure 6a-b, where the within vs. across network correlation strengths for each voxel were averaged across participants in either the matched TD group (6a) or the ASD group (6b). We next identified in the matched TD group the voxels whose correlation within network was stronger than their correlation across networks (those with a positive value in panel a). The distribution of these same voxels in the ASD group is shown in Figure 6c, and these are also the voxels colored red in the scatterplot shown in Figure 6d. For Figures 6e-h the same analysis was repeated, but with the roles of the social orienting network and network 2 reversed. For the estimation of p-values reported in the text which were used to determine whether there was a significant difference between within network and across network correlations, we calculated for each participant the

average within network correlation for each of the two networks, and the average across network correlation, and then carried out a paired two-tailed t-test across participants, on the within network vs. across network correlations. This was done separately for the matched TD group and the ASD group. This is a more conservative estimate than calculating the average within vs. across correlation values for each voxel (averaged across participants), and then carrying out a paired t-test on the within vs. across correlations values across voxels.

31. Data availability: What data exactly will be made public. Is the code for repeating the analyses also included?

We will share the raw fMRI data as well as the eye tracking data, and will upload the unthresholded maps to neurovault. These will be made public upon publication of the manuscript. All code will be made available upon request. We have added this information to the data availability statement (p. 28):

Data are available through NIH Figshare (10.35092/yhjc.c.4741556). All code will be available upon request. Unthresholded maps for the images depicted in Figures 3, 4, 5 and 7, as well as Supplementary Figures 1,2 and 4 are available through Neurovault: <https://identifiers.org/neurovault.collection:6090>

32. Supplementary figure 1 title: “ Correlations between neural typicality and eye movement typicality using across group definitions”
What does “using across group definitions” mean?

Across group definitions means that we are using the network as defined by the TD group to test the ASD group. This title might be confusing though, so we have removed across group definitions from the title, and simply explain the analysis in the legend itself:

Correlation between eye movement typicality and neural typicality for the ASD group (red), averaged across the social orienting network as defined by the correlations of eye movement and neural typicality for the TD group.

Reviewer #2 (Remarks to the Author):

The research was aimed at studying brain regions responsible for social interactions. For this purpose, a group of people with typical development and with Autism Spectrum Disorder were engaged.

The studies were based on an experiment in which selected videos were used for exploring typicality of eye movement and neural responses. The collected data was studied, taking various aspects into account. The analysis included the assessment of the obtained results in terms of their correlations in regard to brain regions and eye

movements.

However, the presentation of the eye-tracking data processing is too general. The method for selecting fixations and scan-paths comparison should be described better. Were scan-paths built from fixation or each registered sample? Please provide an example for calculating a distance between scan-paths.

The data was down sampled to the frame rate at which the movies were presented, and the eye position for each frame was taken to be the average recorded position, we did not identify individual fixations, as this was not such a fine grained analysis. The code for carrying out this analysis will be made available to anyone wishing to replicate this analysis.

The studies can be interesting for prospective readers and can be further developed. However, in some parts, the text is a little bit difficult to read. There are some long sentences, including additional information in brackets, which makes contents understanding demanding.

We thank the reviewer for pointing out this difficulty. We have attempted to improve the readability of the manuscript.

Reviewer #3 (Remarks to the Author):

Reviewers: Mikko Sams and Enrico Glerean

The authors studied the components of the social brain using neurotypical and individuals with autism. They measured eye movement and neural typicality, when participants were viewing Hollywood movie clips depicting social interactions. Brain activity was studied by inter-subject correlation, a measure of similarity and dissimilarity of brain activity of different individuals. Importantly, typicality of eye movements was studied with the different material as used in the fMRI experiment. Typicality of eye movements was compared with typicality of neural activity. Another approach to the data was direct comparison of group neural similarity of neurotypical individuals and individuals with autism. The results revealed two component networks of the social brain, “social orienting” and “inferring mental states of others”.

I especially appreciate the eye movement part of this MS. I tend to think that the MS would be much clearer and useful for the readers, if it would contain only this part: typicality of eye movements and neural activity, and how they are related.

We thank the reviewer for the detailed review. We did in fact consider including only the eye movement results for a simpler manuscript, but we believe that the neural typicality group differences complete the story in a way that is not captured by the eye movements analyses alone. We hope that the clarifications we have made to the manuscript following these reviews have made the points we were trying to get across clearer and easier to follow.

Abstract

This could be improved, it should be made clear how these two networks were found. Perhaps there is not enough space in the Abstract, but then in introduction it should be made clear if these networks were previously known, and now the authors introduce a new method for teasing them apart.

We thank the reviewer for pointing out the conceptual difficulty in the abstract and introduction. While there have been numerous studies on these behaviors, as well as some studies looking at their neural correlates, these two networks have never been fully described before as networks as such. This is partly due to the complex behaviors they underlie, and the difficulty in experimentally probing such complex behaviors. The strength of this paper is in that it combines two large groups of participants from different populations, allowing us to compare responses across populations, with complex and ecologically valid stimuli (social movies), and a new measure of eye movement typicality which can take into account far more complex social orienting behavior than previous studies. We have made some substantial additions and changes to the introduction, which we think now better explain the framework for our work.

Introduction

Give a specific and clear definition to Social orienting network.

As mentioned in reply to the previous point, the social orienting network is not something that has been experimentally defined before as a specific network, though there have of course been studies that have looked at simplistic aspects of this behavior (such as gaze following), and have identified brain regions associated with them. To address this confusion of the terms, we have added the following to the introduction, to clarify that social orienting network is a term that we define here, to correspond to the results of our eye movement analysis (addition in red, p. 2):

*Similarly, while many previous studies have used movies to study the behavioral aspects of social orienting³⁷⁻³⁹, the search for the neural correlates of social orienting has so far utilized only very simple, mostly static and schematic, social stimuli. Moreover, these studies have focused on probing gaze following, which is only one aspect of social cues⁹. These limitations have led to a partial, fragmented understanding of the **neural structures underlying complex social orienting behavior, which we will term the social orienting network.***

Results

As mentioned by the authors, one limitation of the study is that quite important some top and bottom parts of the brain could not be scanned. I take this study more as a methodological one, so in my opinion this is not a crucial limitation. Anyhow, the authors should lengthen their discussion on this issue. What have others found in these areas,

which could be relevant for this study? ISC is not an acronym for average neural typicality.

While we cannot rule out that there are important regions not covered as a limitation of our scan parameters, the regions we are missing are mostly motor, and while they may be important for eye movement control, there is no evidence in the literature linking them directly to social orienting per se. We have added this to our discussion p. 22 (in red):

Please note however that we do not have full coverage of the brain, and some areas which might also be involved in social orienting, most notably frontal and supplementary eye fields, are missing from our analysis (Figure 3). These regions are directly involved in the control of eye movements, and it would therefore be difficult to decouple their role in eye movements from their role in social orienting. Other areas where we are missing coverage are mostly motor, and are not implicated in either social orienting or mentalizing based on previous literature.

Mark the anatomical areas you are talking about in the pictures, it is important to make reading of the figures easy.

We thank the reviewer for this suggestion. We have attempted to add labels to the figures, but given the scale of the figures the labels are either unreadable, or obscure the findings. Since we are describing well known anatomical, rather than functional regions, we hope the readers will be able to navigate these without the labels.

Show neural typicality maps also for the ASD group (cf. Figure 3)

We have added this as a supplementary figure (Supplementary Figure 2) to the paper:

Do I now understand this correctly? The “social orienting network” is the one, which can be found both in TD and ASD participants, and which covaries with the typicality of eye movement patterns. In some parts of this network, neural similarity of TD and ASD participants differ. The second network is found by directly combining neural typicality of TD and ASD subject, involving those areas where there. This is what has been done before in previous studies, so this analysis should confirm and perhaps extend results obtained in them.

Indeed, that is correct. We have made several changes to the manuscript which now better explain this point, and the distinction between the networks. The social orienting network is defined by the covariation with eye movement typicality. Figure 4b in the original submission shows this network in the combined TD and ASD group. We have added a supplementary figure which directly examines this network in the ASD group separately from the TD group (Supplementary Figure 1a). The second network is defined by directly comparing the neural typicality of the TD and ASD groups. Though this has been done before in three previous studies, those have mostly been very small and have had very inconsistent results, with two studies showing conflicting results, and the third study not giving the locations of these group differences. Our results agree with the results from one of the two studies, and in that sense confirm and extend this literature. Since there might be an interaction with these group differences in typicality in

the effects of the correlations with eye movements, we have also added an analysis of the combined TD/ASD group after removing the group mean of both the eye movement typicality and the neural typicality from each group, so that we can better understand what part of this network in the combined group is driven by the group differences in neural typicality. This is shown in the new Supplementary Figure 1b.

The part of the MS analyzing the differences of these two networks is not very easy to read. Perhaps it would be better to start with the comparison of neural similarity between TD and ASD participants, as done previously. And then going to the analysis using eye movements, as a new behavioural method, which then shows something new, a social orienting network. This network then appears to be quite similar in TD and ASD participants, even though within it neural similarity is smaller in ASD participants. But there is additional difference found in comparing neural similarity, which is not correlating with eye movements. This then is suggested to be related to mentalizing.

I have difficulties in following the reasoning here. It may be correct, but should be written more clearly for a basic human neuroscientist to follow the story and be convinced of the reasoning. So my suggestion is to spend more time on this, also being more economical in writing.

We thanks the reviewer for this suggestion, we have in fact previously attempted to write the manuscript in this order, but found it to be more cumbersome and difficult to explain. We have added another figure to the results, which is now Figure 6, which we hope clarifies the connectivity analysis which is the basis for the difference in the distinct sub-networks which are portrayed in the last figure (now Figure 7).

We have also substantially reworked the discussion, to better explain the framework for our results. The first two paragraphs of the discussion now put these results in much clearer context, and read as follows (pp. 19-20, additions / changes in red):

Movie viewing evokes both shared neural responses, and shared behavior, in the form of eye movements orchestrated by carefully constructed visual, auditory and social cues. Yet there is individual variation in both behavior and neural responses. Evidence from previous studies suggests that neural typicality, or correlations between subjects in their neural responses, indicate shared processing, or a shared experience relevant to the function of the correlated region. For example, studies have demonstrated that for participants viewing a movie, or listening to a narrative, the similarity of their interpretation of the story predicts the degree of neural similarity in regions involved in narrative interpretation, such as the fronto-parietal network and the default mode network²⁰. Likewise, participants recalling the same events have been shown to have more neural similarity than participants recalling different events¹⁹. ASD participants have difficulties in both social orienting, and higher order mentalizing. These are likely the result of different processes, as previous studies have demonstrated a decoupling of high order comprehension / mentalizing during movie viewing from eye movement patterns¹⁵. Using our measures of the correlation between eye movement typicality and neural typicality, and group differences between the TD and ASD groups in neural typicality, we therefore expected to find two different, partially overlapping networks. The correlation between the two typicality measures across participants was used to search for the network responsible for social orienting. As the neural typicality measure is an indicator of shared processing, we expected lower neural typicality in the ASD group in regions involved in the processes in which ASD participants differ from their TD counterparts, namely social orienting and higher order social comprehension / mentalizing. Therefore, we expected to find group differences in neural typicality in the regions underlying these behaviors.

For the social orienting network, on top of previous research²⁷⁻²⁹, our own data described here shows reduced typicality of eye movements in the ASD group compared with the TD group (Figure 2), and this should be reflected in the typicality of neural processing. However, at the

single voxel level, we did not find group differences in neural typicality in most voxels of the social orienting network. Those differences were present at the network level though, as there was significantly lower neural typicality in the ASD group within the social orienting network as a whole, perhaps indicating a lack of sensitivity of the neural typicality group difference measure. The more robust effects, which could be seen at the single voxel level (Figure 5) centered on regions which seemed to constitute a separate, distinct network, whose correspondence to the mentalizing network will be discussed below.

Methods

Give the details of the film moving during eye tracking in enough details that the experiment could be replicated.

We thank the reviewer for this comment. We have attempted to provide as much information as possible about these movies in the methods (p.25):

These movie clips were all taken from Hollywood movies – The Blind Side (6 clips), Goonies (4 clips), How To Lose a Guy in Ten Days (4 clips), The Italian Job (5 clips), and The Neverending Story (5 clips) and were all 14 seconds long. There was ongoing dialogue between at least two characters in all these scenes, though characters were often not on the screen together.

Due to the different version of these movies which can be found in different sites, we are unsure how we could better identify these movie clips, as giving a start time from the beginning might be misleading. Though publicly posting these movie clips might be a violation of copyright, we are happy to share all movie clips used in this experiment upon request.

Please give the IQs and other this type of information of the subjects.

We have added the IQ scores to the participants section of the methods:

mean IQ for the ASD participants was 108.7, std = 13.8; mean IQ for the matched TD participants was 110, std = 13.3

It is not clear how the intersubject correlation was computed: was it done pairwise (as recommended by previous works of Gang CHen or Jussi Tohka), or is it that the time series of a subject is correlated to the average of all other N-1 subjects? From the manuscript it seems that the time series of a subject is correlated with the average across all subjects (including the subject of interest), if this is the case then the statistics will be biased as there is always a non zero correlation. Opening the code (see below) would increase the transparency of the methods. Similar concerns apply to the measure of typicality for eye tracking: was it a correlation between an individual time series with the average of all participant' time series (including the subject of interest)? These procedures should be clarified in the methods section.

We thank the reviewer for raising this point. Both measures of typicality were calculated by comparing the subject of interest to the average of all other subjects, not including himself. This is now clarified in the data analysis part of the methods section (pp.26-27):

Eye movement typicality for each of the TD participants was calculated per movie per frame, by computing the Euclidean distance of their average recorded eye position per frame to the mean recorded eye position of all other participants (i.e. all N-1 participants, not including the participant of interest) for each frame. These distances were then averaged for each participant across all frames of all movies, to obtain one typicality measure. Neural typicality was similarly assessed per participant, per voxel, by calculating the correlation of the time course of each voxel for that participant to the average time course of that voxel for all the other (N-1) participants. The correlation between the eye movement typicality and the neural typicality was calculated based on these two measures.

Correlation between eye movement typicality and neural typicality: is this done across pairs (i.e. mantel test) so that the correlation value of a subject pair brain time series is compared to a correlation value of their eye tracking time series? Or is this done across individual subjects? Please clarify in the methods section.

This measure was calculated using the typicality values calculated from the average scan path / BOLD time course of all other (N-1) subjects, as we hope is now clear from the above description in the methods. We chose not to use the pairwise correlations, as unlike some of the other papers using the pairwise method our goal was to assess “typicality”, which by definition is in relation to the average, rather than calculate distances between individual participants. However, as a sanity check, we also carried out a pairwise calculation of both the eye movement typicality and the neural typicality, and examined the similarity between the values obtained using the two methods. We compared the correlation (across participants, for each voxel) between the average pairwise eye movement typicality for each participant to the average pairwise neural typicality of each participant, to the correlation obtained using the correlation to the average of all other N-1 participants. The two measures were very highly correlated across voxels ($r = 0.84$).

It is not clear if the p and q values used in figure 5 were derived from the t-statistics or if they were computed with permutation-based clustering. Since in the methods you specify that all correction for multiple comparisons were done with permutation-based clustering, the same approach should also be used in figure 5.

We thank the reviewer for pointing out this discrepancy. We have redone this analysis using the cluster size correction

Please upload all unthresholded statistical brain maps to NeuroVault and include the neurovault link into the manuscript. NeuroVault is a tool developed at Stanford to share result maps, this enables future meta-analyses as well as allowing the readers to

explore the results interactively. Neurovault collection is then automatically linked to the paper after the paper is published, increasing its visibility.

We have created a placeholder in Neurovault for the unthresholded maps, and we will make these public these upon publication of the manuscript.

Code: you mentioned in-house software written in MATLAB. While this is not mandatory, for transparency it is highly recommended to share the code used. Even if the code is not perfect, or optimised, or documented, it is still better than having no code at all.

We are committed to the principles of open science, and will gladly share all code upon request. We have added this information to the Data Availability statement

It seems that data can be shared, which is fantastic. Please include already the link to the data repository in the manuscript to make the data FAIR.

We will share all datasets through NIH Figshare, and will make all data publicly available upon publication. We have added the relevant links to the Data Availability section:

*Data are available through NIH Figshare (10.35092/yhjc.c.4741556). All code will be available upon request. Unthresholded maps for the images depicted in Figures 3, 4, 5 and 7, as well as Supplementary Figures 1,2 and 4 are available through Neurovault:
<https://identifiers.org/neurovault.collection:6090>*

Perhaps it would be more sensitive to use non-Hollywood clips to have more natural differences in gaze patterns? Please discuss.

In this study, we purposely chose not only Hollywood clips, but clips which were the most likely to result in typical gaze patterns. The reasoning for this was that in order for typicality to be a useful measure, there must be a typical, or “ideal” scan path, otherwise we are comparing to noise. The use of multiple clips was designed to allow for greater sensitivity, as small, consistent differences in typicality could be seen in the averages.

REVIEWERS' COMMENTS:

Reviewer #1 (Remarks to the Author):

The authors have done a thorough revision and generally addressed my concerns and also corrected a small error in an initial analysis, which did not substantially affect the results or the conclusions of the study. Some small comments follow, which should be easy to address before publication.

In response to point 9 (how was the significance calculated for eye movement vs. brain similarity), the authors responded "Significance in this case was determined by calculating the p-value of the correlation coefficient". However, this does not actually reveal the test they used to get to the p-value, it is rather only a definition of significance. One way probability of correlation is sometimes defined is by converting r-values to t-values and reading the p-value from the t-distribution with the appropriate degrees of freedom, which is perhaps what the authors mean since they refer to sample sizes. However, since elsewhere in the manuscript the authors use non-parametric permutation tests, it would seem logical to also use non-parametric statistics here, particularly since the assumption about independent samples in the sample is not strictly followed because each participant is part of the model of the other participants. This is not likely to seriously affect the results.

For the figures showing multiple results in one panel (Fig 4b, Fig 7 and Fig S4 at least, potentially scatter plots in Fig.6) it would be helpful to still add a legend showing what each of the colors mean. It would greatly help in understanding at a glance what the figure depicts.

Additionally, in a response to another reviewers' comments, the author suggest that the parietal and precentral regions that are not covered by the data include mainly areas relevant for eye movement control and motor execution. However, some parietal regions along the intraparietal sulcus together with frontal eye fields seem also important for spatial attention and may be particularly strongly activated during attention to action perception ("action observation network"), so these regions may indeed be relevant for social orienting, though not selectively. While the authors cannot retroactively change the slice locations, they should rephrase their characterization of the regions.

As an additional note, while I concur that there are clearly independent brain regions that show participating in the two networks the authors consider, the analysis separating regions to areas that correlate more strongly within a network than between networks still seems to preclude overlapping regions in some of the figures. The comments in the initial did not mean to suggest that this is equivalent to just excluding overlapping regions, because all non-overlapping regions do not need to satisfy this criterion. However, this is no longer problematic with the current presentation of the results that is less strict about fully non-overlapping networks, as well as the clarification for the figure caption.

Finally, in line with other reviewers, I would also find it helpful to share the code in addition to the data, even if it is not perfectly organized and commented. However, this does not seem to be a requirement.

I recommend the article for publication when the authors have considered these minor points.

Response to reviewer

The authors have done a thorough revision and generally addressed my concerns and also corrected a small error in an initial analysis, which did not substantially affect the results or the conclusions of the study. Some small comments follow, which should be easy to address before publication.

In response to point 9 (how was the significance calculated for eye movement vs. brain similarity), the authors responded “Significance in this case was determined by calculating the p-value of the correlation coefficient”. However, this does not actually reveal the test they used to get to the p-value, it is rather only a definition of significance. One way probability of correlation is sometimes defined is by converting r-values to t-values and reading the p-value from the t-distribution with the appropriate degrees of freedom, which is perhaps what the authors mean since they refer to sample sizes. However, since elsewhere in the manuscript the authors use non-parametric permutation tests, it would seem logical to also use non-parametric statistics here, particularly since the assumption about independent samples in the sample is not strictly followed because each participant is part of the model of the other participants. This is not likely to seriously affect the results.

We thank the reviewer for pointing out that this point was still not made sufficiently clear in the manuscript. We used this parametric test (indeed, converting to t-values and reading the p-value from the t-distribution, which we have now noted in the manuscript, p.11) because this is a voxel-wise test, and a non-parametric test would give us a different value for each voxel, making it difficult to determine the correct threshold. Similarly, for the other voxel-wise test comparing neural typicality in the TD group and the ASD group, we also used a parametric t-test for this same reason. However, given the reviewer’s comment, we also carried out a non-parametric permutation test to determine significance at the voxel level. As mentioned, there are several different ways to determine the correct threshold using this method (using different thresholds for different voxels, or taking the 95% or 99% across all voxels). The overall threshold, taking the 95th percentile for each voxel, and then the 95th percentile across all voxels, was nearly identical to the parametric test at a threshold of $p = 0.05$ down to two decimal points (with the parametric test being slightly more stringent), and similarly the permutation test at the 99th percentile for each voxel was also within two decimal points of the parametric test threshold value. For easier readability, we decided to leave the text with the parametric test.

For the figures showing multiple results in one panel (Fig 4b, Fig 7 and Fig S4 at least, potentially scatter plots in Fig.6) it would be helpful to still add a legend showing what each of the colors mean. It would greatly help in understanding at a glance what the figure depicts.

We thank the reviewer for this suggestion, this indeed substantially improves the ease of interpretation of the results. We have added legends to all the figures.

Additionally, in a response to another reviewers' comments, the author suggest that the parietal and precentral regions that are not covered by the data include mainly areas relevant for eye movement control and motor execution. However, some parietal regions along the intraparietal parietal sulcus together with frontal eye fields seem also important for spatial attention and may be particularly strongly activated during attention to action perception ("action observation network"), so these regions may indeed be relevant for social orienting, though not selectively. While the authors cannot retroactively change the slice locations, they should rephrase their characterization of the regions.

We have added this caveat to our discussion of the limitations imposed by our partial coverage (p. 22, new text in red):

Please note however that we do not have full coverage of the brain, and some areas which might also be involved in social orienting, most notably frontal and supplementary eye fields as well as some parietal regions along the intraparietal sulcus, are missing from our analysis (Figure 3). These regions have previously been implicated in spatial attention tasks, and in particular attention to action perception, making them potentially relevant for social orienting⁴⁸. However, frontal and supplementary eye fields are also directly involved in the control of eye movements, and it would therefore be difficult to decouple their role in eye movements from their role in social orienting.

As an additional note, while I concur that there are clearly independent brain regions that show participating in the two networks the authors consider, the analysis separating regions to areas that correlate more strongly within a network than between networks still seems to preclude overlapping regions in some of the figures. The comments in the initial did not mean to suggest that this is equivalent to just excluding overlapping regions, because all non-overlapping regions do not need to satisfy this criterion. However, this is no longer problematic with the current presentation of the results that is less strict about fully non-overlapping networks, as well as the clarification for the figure caption.

We agree with the reviewer, and are pleased that the changes we have made to the presentation of the results have clarified this point.

Finally, in line with other reviewers, I would also find it helpful to share the code in addition to the data, even if it is not perfectly organized and commented. However, this does not seem to be a requirement.

We are indeed very happy to make the full code available upon request, as is reflected in the data availability statement.

I recommend the article for publication when the authors have considered these minor points.

We would like to thank the reviewer once again for the thorough and thoughtful review.